# [RE] GNNBoundary: Finding Boundaries and Going Beyond Them

**Jan Henrik Bertrand** *                    *jan.henrik.bertrand@student.uva.nl*
*University of Amsterdam*

**Lukas Bierling** *                          *lukas.bierling@student.uva.nl*
*University of Amsterdam*

**Ina Klaric** *                              *ina.klaric@student.uva.nl*
*University of Amsterdam*

**Aron Wezenberg**                            *aron.wezenberg.wezenberg@student.uva.nl*
*University of Amsterdam*

**Reviewed on OpenReview:** *https://openreview.net/forum?id=XSgCOIMqDO*

## Abstract

Graph classification models are becoming increasingly popular, while explainability methods face challenges due to the discrete nature of graphs and other factors. However, investigating model decision making, such as through decision-boundary regions, helps prevent misclassifications and improve model robustness. This study aims to reproduce the findings of *GNNBoundary: Towards Explaining Graph Neural Networks Through the Lens of Decision Boundaries* (Wang & Shen, 2024). Their work supports 3 main claims: (1) their proposed algorithm can identify adjacent class pairs reliably, (2) their GNNBoundary can effectively and consistently generate near-boundary graphs outperforming the cross-entropy baseline, and (3) the generated near-boundary graphs can be used to accurately assess key properties of the decision boundary; margin, thickness, and complexity. We reproduced the experiments on the same datasets and extended them to two additional real-world datasets. Beyond that, we test different boundary probability ranges and their effect on decision boundary metrics, develop an additional baseline, and perform hyperparameter tuning. We confirm the first claim regarding the adjacency discovery as well as the second claim that GNNBoundary outperforms the cross-entropy baseline under the limitation that it requires intensive hyperparameter tuning for convergence. The third claim is partially accepted as we observe a high variance between the reported and obtained results, proving the reliability and precision of the boundary statistics. Code and instructions are available at: https://github.com/jhb300/re_gnnboundary.

## 1 Introduction

Reproducibility is the backbone of scientific progress, ensuring that findings are robust, reliable, and generalizable. In that, model explainability methods are often used by researchers to assess and improve the robustness of a model. This study focuses on GNNBoundary by Wang & Shen, a framework designed to analyze decision boundaries in graph neural networks (GNN) (Scarselli et al., 2009), more specifically in GNN-based classifiers. Understanding decision boundaries is crucial for evaluating model robustness, as well as identifying and understanding modes of failure. This is especially important for real-world applications in

---

* Equal contribution.

domains such as social networks, biology, and recommendation systems (Fan et al., 2023). The primary motivation behind this work lies in the originality of the approach: to the best of our knowledge, GNNBoundary represents the first method specifically aimed at investigating decision boundaries in GNNs. Furthermore, previous reproducibility efforts concerning its methodological predecessor, GNNInterpreter (Vasilcoiu et al., 2024), revealed that several key results could not be replicated. This underscores the importance of thoroughly validating GNNBoundary's findings through a dedicated reproducibility study.

GNNs have shown remarkable performance in tasks that involve graph data, such as node classification, link prediction, and graph classification (Errica et al., 2020). Despite their widespread utilization, the complexity of these models often leads to challenges in understanding their decision-making. GNNBoundary, as a post hoc method, attempts to open this black-box by investigating decision boundaries with quantitative metrics and visualizing near-boundary graphs for adjacent class pairs. In doing so, the method offers insight into how GNNs distinguish between classes and the robustness of the classification.

This reproducibility study aims to verify the results and claims made in the original GNNBoundary paper by Wang & Shen (2024). Specifically, this study attempts to determine whether the GNNBoundary framework can be used to consistently and reliably generate near-boundary graphs and collect boundary metrics for the discovered adjacent class pairs. This involves reproducing the authors' experiments, verifying the correctness of the implementation, and evaluating the robustness of the findings across four datasets. Motif (Wang & Shen (2023)), Collab (Yanardag & Vishwanathan (2015)), Enzymes (Schomburg et al. (2004)), and IMDB (Yanardag & Vishwanathan (2015)).

Beyond reproducing the results from Wang & Shen (2024) using their instructions, our objective is to enhance the applicability of GNNBoundary to practitioners and researchers by providing additional guidance. Moreover, we provide insights on the generalizability and robustness of the method through additional experiments and analyses.

## 2 Reproducibility Considerations

To guide our efforts, the claims and contributions defined in this section will carry through the following sections. The main claims made by the authors are as follows:

**Main Claims:**

1. Their proposed algorithm for identifying adjacent class pairs can reliably identify the degree of adjacency of a class pair.

2. GNNBoundary can effectively and consistently generate near-boundary graphs with faster convergence and a higher success rate than the cross-entropy baseline.

3. The generated near-boundary graphs can be used to accurately assess key properties of the decision boundary: margin, thickness, and complexity.

Our main contributions beyond the reproduction of the results from Wang & Shen (2024) are:

**Contributions:**

1. Performing hyperparameter optimization to systematically analyze the impact of different configurations on convergence behavior, leading to faster and more stable model training.

2. Developing an additional baseline by using over 500 GNNInterpreter graphs (Wang & Shen, 2023) per class in each dataset and connecting them via randomly assigned edges.

3. Extending experiments with two real-world datasets: posts from the forum site Reddit and a relational graph dataset IMDB, with actors and actresses connected by co-appearances in movies.

4. Investigating the trade-off between boundary metric approximation quality and target class probability ranges (e.g., $[0.49, 0.51]$, $[0.45, 0.55]$), offering practical insights into the training requirements for near-boundary graph sampling.

## 3 Background

### 3.1 Graph Neural networks

An increasing amount of graph structured data creates a need for a graph structure learning systems. Graph Neural Networks (GNNs) emerge from its predecessors in deep learning as models to operate on graph structures. Graphs have unique characteristics that differentiate them from other data types, such as a variable number of nodes and edges across graphs. This variability complicates the application of convolutions. Moreover, unlike images or text, where individual data instances are typically independent, graphs exhibit inherent dependencies, as nodes can be interconnected and therefore influence each other (Wu et al., 2020). GNNs are networks that aim to create a representation vector of a node or a whole graph given the features of the node and the graph structure (Xu et al., 2019). The higher need for graph analysis and the emergence of various GNNs such as Graph Convolutional Networks (Kipf & Welling, 2017) and Graph Attention Networks (Veličković et al., 2018), create the need for explainability and interpretability in the GNN domain.

### 3.2 Explainability of GNNs

In a review of explainability approaches to GNNs by Kakkad et al. the explanation is described as either self-interpretability or post-hoc explainability methods. The former enables the model to justify its predictions during training and influence it, while the latter considers the training as a black-box process that is to be explained post-training. GNNBoundary is a post-hoc explainability method. It examines the decision boundaries of a trained GNN by learning to generate graphs near the boundary. The final distinction in the review is between instance-level and model-level explainability. Instance-level explainability aims to explain predictions of a certain instance, while the model-level method aims to justify the high-level decision-making process of the whole model. GNNBoundary is considered a model-level method as the generation of near-boundary graphs aims to explain global patterns in the decision process of a GNN classifier. In literature previous to GNNBoundary, generation-based methods do not yet explore the decision boundaries. However some methods explore the model-level behavior by generating graphs representative of a class, such as GNNInterpreter (Wang & Shen, 2023) and XGNN (Yuan et al., 2020).

### 3.3 Decision region and boundary

A classifier $f$ partitions the $d$-dimensional space $\mathbb{R}^d$ into $C$ decision regions $\mathcal{R}_1, \mathcal{R}_2, \ldots, \mathcal{R}_C$, such that for any $G \in \mathcal{R}_c$, the predicted class is $c = argmax_k f_k(G)$ where $k \in [1, C]$ (Karimi et al., 2019). The decision boundary between class $c_1$ and class $c_2$ is defined as $\mathcal{B}_{c_1 \| c_2} = \{G : f_{c_1}(G) = f_{c_2}(G) > f_{c'}(G), \forall c' \neq c_1, c_2\}$, representing the set of graphs $G$ where the classifier assigns equal probability to classes $c_1$ and $c_2$, while ranking them higher than all other classes. For the embedding space, Wang & Shen (GNNBoundary) define

$$\mathcal{B}_{c_1 \| c_2}^{(l)} = \{\mathbf{H}^{(l)} : \sigma(\eta_l(\mathbf{H}^{(l)}))_{c_1} = \sigma(\eta_l(\mathbf{H}^{(l)}))_{c_2} > \sigma(\eta_l(\mathbf{H}^{(l)}))_{c'}, \forall c' \neq c_1, c_2\} \tag{1}$$

where $\sigma$ is the Softmax activation function and $\eta_l$ are the last $L - l$ layers of the discriminator $f$ that output the logits for graph embedding $\mathbf{H}^{(l)}$ after layer $l$. In other words, the decision boundary represents the points in $\mathbb{R}^d$ where the classifier is "uncertain" about assigning a label between classes $c_1$ and $c_2$.

## 4 GNNBoundary

Wang & Shen (2024) propose an adjacency finding algorithm, an objective function to generate near-boundary graphs for adjacent class pairs and a dynamic regularization scheduler to avoid local minima.

**Notation.** A graph is represented as $G = (\mathcal{V}, \mathcal{E})$, where $\mathcal{V} = \{v_1, v_2, \ldots, v_N\}$ is the set of nodes and $\mathcal{E} \subseteq \mathcal{V} \times \mathcal{V}$ is the set of edges. The total number of nodes is denoted by $N$, and the total number of edges is denoted by $M$. The adjacency relationships between nodes are captured by the adjacency matrix $\mathbf{A} \in \{0, 1\}^{N \times N}$, where an entry $a_{ij} = 1$ indicates the presence of an edge between node $v_i$ and node $v_j$, while $a_{ij} = 0$ indicates the absence of an edge. Node features are stored in the feature matrix $\mathbf{Z} \in \mathbb{R}^{N \times d}$, where

$\mathbf{z}_i \in \mathbb{R}^d$ represents the feature vector of node $v_i$ and therefore the $i$-th row of $\mathbf{Z}$. Each class $c$ is associated with a decision region $\mathcal{R}_c$, which represents the set of points in the feature space that the classifier assigns to class $c$.

## 4.1 Adjacency Discovery

Running the boundary analysis for all class pairs would be computationally prohibitive. Hence, Wang & Shen propose an adjacency discovery method that calculates adjacency rates based on the dataset and all but the last layer of discriminator model $f$ denoted as embedding function $\eta_{L-1}$. This comes with the advantage that the embedding space after the last hidden layer has linear decision boundaries. The algorithm samples graphs $G_{c_1} \in \mathcal{R}_{c_1}$ and $G_{c_2} \in \mathcal{R}_{c_2}$, embeds them using $\eta_{L-1}$ obtaining $\mathbf{H}_{c_1}^{(L-1)}$ and $\mathbf{H}_{c_2}^{(L-1)}$ respectively. They then interpolate between $\mathbf{H}_{c_1}^{(L-1)}$ and $\mathbf{H}_{c_2}^{(L-1)}$ in the embedding space and use the remaining layer of the discriminator $f$ to determine if the interpolated embedding is part of any intermediate decision region other than $\mathcal{R}_{c_1}$ or $\mathcal{R}_{c_2}$. The final adjacency score is the share of the $K$ sampled graph pairs that do not cross any decision region other than $\mathcal{R}_{c_1}$ or $\mathcal{R}_{c_2}$.

## 4.2 Graph Generation

**GNNBoundary Sampling.** As discrete structures, graphs are not inherently differentiable. However, Wang & Shen propose a relaxation approach to mitigate this issue. The discrete graph structure is relaxed into a differentiable form allowing gradient-based optimization, using the reparameterization trick inspired by Jang et al. (2017), Wang & Shen (2023) and Luo et al. (2020). The boundary graphs are modeled as Gilbert random graphs (Gilbert, 1959), where the probability distribution of the graph $P(G)$ is formulated as the product of node feature probabilities $P(z_i)$ and edge probabilities $P(a_{ij})$. Edges are sampled from a Bernoulli distribution, while node features follow a Categorical distribution. To enable gradient-based learning, the categorical variables are relaxed using the Concrete distribution (Maddison et al., 2017), leading to a differentiable approximation. The concrete distribution is a continuous version of the Categorical distribution with closed-form density, $\tilde{z}_i \sim \text{Concrete}(\zeta_i, \tau_z)$ for the node features $\tilde{a}_{ij} \sim \text{BinaryConcrete}(\omega_{ij}, \tau_a)$ for edges, where $\tau_z$ and $\tau_a$ are hyperparameters to control the approximation of Categorical distribution, $\omega_{ij} \in \Omega$ and $\zeta_i \in \mathcal{Z}$. $\Omega$ and $\mathcal{Z}$ denote the set of all edge and node feature logits, respectively, represented as matrices in practice. For sampling, the authors of GNNBoundary (Wang & Shen, 2024) utilize the Gumbel-Softmax trick (Jang et al., 2017), which ensures that both the edge variables and node feature variables remain differentiable. This is achieved by computing edge variables $\tilde{a}_{ij}$ and node feature variables $\tilde{z}_i$ using the transformations $\tilde{a}_{ij} = \text{Sigmoid}\left((\omega_{ij} + \log \epsilon - \log(1-\epsilon))/\tau_a\right)$ and $\tilde{z}_i = \text{Softmax}((\zeta_i - \log(-\log \epsilon))/\tau_z)$. Furthermore, the graph distribution is learned by minimizing the expected loss function through Monte Carlo sampling and gradient descent, making the boundary graph generation process efficient for discrete graph structures, and as follows:

$$\min_{\mathbf{A},\mathbf{Z}} \mathcal{L}(G) = \min_{\mathbf{\Theta},\mathbf{P}} \mathbb{E}_{G \sim P(G)}\left[\mathcal{L}(\mathbf{A}, \mathbf{Z})\right] \approx \min_{\mathbf{\Omega},\mathbf{Z}} \mathbb{E}_{\epsilon \sim U(0,1)}\left[\mathcal{L}(\tilde{\mathbf{A}}, \tilde{\mathbf{Z}})\right] \approx \min_{\mathbf{\Omega},\mathbf{Z}} \frac{1}{K}\sum_{k=1}^{K} \mathcal{L}(\tilde{\mathbf{A}}, \tilde{\mathbf{Z}}). \tag{2}$$

where $\mathbf{A}$ and $\mathbf{Z}$ denote the adjacency matrix and node feature matrix of the graph. We point out that this sampling framework builds on a strong independence assumption. Specifically, the probability distributions for the edges, nodes and node features are all independent of each other in this framework. While this assumption is needed to make the sampling computationally tractable, it yields a potentially inaccurate model of the graph sampling space. Future research could work on ways to mitigate this issue.

**Boundary Criterion.** To generate boundary graphs, the probability of a boundary graph belonging to both classes should be equal, i.e. $G_{c_1 \| c_2} \in \mathcal{B}_{c_1 \| c_2}$ with $\sigma(f(G))_{c_1} = \sigma(f(G))_{c_2} = 0.5$. As this is usually unattainable in practice, the authors propose a relaxed class probability range that the boundary graphs are permitted to belong to. Thus, the stopping criterion for the optimization that determines whether a graph $G$ is close enough to the boundary $\mathcal{B}_{c_1 \| c_2}$ is defined as:

$$\Psi(G) = \mathbb{1}_{p(c_1), p(c_2) \in [p_{\min}, p_{\max}]}(G). \tag{3}$$

The authors suggest using $p_{min} = 0.45$ and $p_{max} = 0.55$. Due to this relaxation, the term "near-boundary graph" is more accurate than "boundary graph" and will be used in the following sections.

**Optimization.** To generate near-boundary graphs, the optimization objective must be designed to balance the trade-off between boundary classes while ensuring efficiency. A key limitation of using the cross-entropy loss as an objective function is its inability to fully satisfy the required constraints for near-boundary graph generation, as it still may lead to the minimization of one of the logit values of the boundary classes. This does not serve the goal of producing near-boundary graphs that have features of both adjacent classes. An improved objective function is proposed that encourages posterior probabilities for boundary classes, $p(c_1) = p(c_2) = 0.5$, while minimizing unwanted class probabilities, $b' \notin \{c_1, c_2\}$. Additionally, a squared penalty is introduced that penalizes logit values that differ from the target class probability vector. This optimization technique is essentially an "enhanced" cross-entropy that encourages higher probabilities for both adjacent classes and penalizes deviated logits. Formally,

$$\min_G \mathcal{L}(G) = \min_G \sum_{b' \notin \{c_1, c_2\}} \beta f(G)_{b'} \cdot p^*(b')^2 - \sum_{b \in \{c_1, c_2\}} \alpha f(G)_b \cdot (1 - p^*(b))^2 \cdot \mathbb{1}_{p^*(b) < \max_{c \in [1,C]} p^*(c)}, \qquad (4)$$

where $f(G)_{c_1}$ and $f(G)_{c_2}$ are logits of the function $f(G)$ of the two adjacent classes, $p^*(c)$ is the detached version, or no gradient version, of the posterior probability $p(c) = \sigma(f(G))_c$ and $\alpha$ and $\beta$ are constant hyperparameters.

**Regularization.** To encourage the interpretability of the generated graphs to be as succinct as possible, Wang & Shen impose a budget penalty to limit the size of the explanation graph in terms of the number of nodes and edges. Concretely, they propose the use of $L_1$ and $L_2$ regularization:

$$R_{\text{budget}} = \text{Softplus}\left(\|\text{Sigmoid}(\mathbf{\Omega})\|_1 - B\right)^2, \qquad (5)$$

with B being the expected maximum number of nodes in a boundary graph G. This regularization penalizes the model when the expected number of edges, computed from the relaxed adjacency matrix, exceeds a predefined budget. Given that certain patterns require certain graph sizes, a size penalty is a potential problem for convergence. Hence, the authors propose a dynamic scheduling method within the training procedure that adapts the budget penalty. To not hinder convergence, a smaller penalty is applied on graphs further away from the decision boundary and bigger penalty on graphs closer to the decision boundary. The budget penalty weight is defined as,

$$w_{\text{budget}}^{(t)} = w_{\text{budget}}^{(t-1)} \cdot s_{\text{inc}}^{\mathbb{1}\{\Psi(G^{(t)})\}} \cdot s_{\text{dec}}^{\mathbb{1}\{\neg\Psi(G^{(t)}) \wedge (s_{\text{dec}} \cdot w_{\text{budget}}^{(t-1)} \geq w_{\text{budget}}^{(0)})\}}, \qquad (6)$$

where $w_{budget}^{(0)}$ is a hyperparameter for the initial weight, $s_{inc}$ is for weight increment, $s_{dec}$ is for weight decrement and $G^{(t)} = \mathbb{E}_{G \sim P(G)}[G]$ for an optimization iteration $t$. The dynamic regularization can help convergence by permitting the budget penalty to interfere with the main loss function.

## 4.3 Boundary Analysis

**Margin.** The boundary margin quantifies the minimum separation between decision regions in a graph classification model (Yang et al., 2020). Unlike classical margin definitions, which are based on worst-case distances, Wang & Shen (2024) use class representative graphs (also denoted as $G_{c_1}$ in the following) generated using the GNNInterpreter framework (Wang & Shen, 2023). Given a dataset $\mathcal{D}$, a classification function $f$, and graph representations $G_{c_1}$ and $G_{c_1\|c_2}$ belonging to different decision regions, the boundary margin is defined as:

$$\Phi(f, c_1, c_2) = \min_{(G_{c_1}, G_{c_1\|c_2})} \|\phi_l(G_{c_1}) - \phi_l(G_{c_1\|c_2})\| \qquad (7)$$

where $G_{c_1} \in \mathcal{R}_{c_1}$ and $G_{c_1\|c_2} \in \mathcal{B}_{c_1}$ represent graph samples from class $c_1$ and its nearest boundary region with class $c_2$, respectively. The function $\phi_l(G)$ denotes the graph embedding function extracted from classifier $f$, mapping the input graphs into a learned feature space. A larger margin implies better class separation, contributing to model robustness, while a smaller margin suggests higher decision boundary instability, increasing the risk of misclassification (Yang et al., 2020).

**Thickness.** Boundary thickness is a metric that quantifies the width of a decision boundary. Given a classification function $f$, the asymmetric boundary thickness $\Theta(f, \gamma, c_1, c_2)$ is defined as the expected distance between pairs of graphs $G_{c_1}$ and $G_{c_1\|c_2}$, sampled from a distribution $P$, weighted by the fraction of a continuous interpolation between these graphs where the posterior difference satisfies a margin condition. Formally, this is expressed as:

$$\Theta(f, \gamma, c_1, c_2) = \mathbb{E}_{(G_{c_1}, G_{c_1\|c_2}) \sim P} \left[ \left\| \phi_l(G_{c_1}) - \phi_l(G_{c_1\|c_2}) \right\| \int_0^1 \mathbb{1}_{\gamma > \sigma(\eta_l(h(t)))_{c_1} - \sigma(\eta_l(h(t)))_{c_2}} dt \right] \tag{8}$$

where $h(t) = (1-t) \cdot \phi_l(G_{c_1}) + t \cdot \phi_l(G_{c_1\|c_2})$ defines an interpolation between the boundary graph embedding $\phi_l(G_{c_1\|c_2})$ and class graph embedding $\phi_l(G_{c_1})$. $\gamma$ is a threshold parameter that defines a margin condition for evaluating the boundary thickness, it is set at 0.75 to capture high uncertainty regions. Boundary thickness captures the average width of the decision boundary. If the decision boundary has a low width and thus separates the data well, the integral in eq. (8) will become a small number. Hence, a low boundary thickness corresponds to a low graph density around the decision boundary and with that good separation of the data.

**Complexity.** The boundary complexity measure calculates the structural complexity of a classifier's decision boundary by analyzing the distribution of adversarial examples in feature space. Using Principal Component Analysis (PCA), the spread of adversarial examples is captured through the eigenvalues $\boldsymbol{\lambda}$ of their covariance matrix (Guan & Loew (2020)). A simpler decision boundary results in adversarial examples aligning along a single eigenvector, whereas a more complex boundary distributes them across multiple eigenvectors. This complexity is computed as the Shannon entropy of the normalized eigenvalues:

$$\Gamma(f, c_1, c_2) = H\left( \frac{\boldsymbol{\lambda}}{\|\boldsymbol{\lambda}\|_1} \right) / \log D = \left( -\sum_i \frac{\lambda_i}{\|\boldsymbol{\lambda}\|_1} \log\left( \frac{\lambda_i}{\|\boldsymbol{\lambda}\|_1} \right) \right) / \log D. \tag{9}$$

Here, $\boldsymbol{\lambda} = (\lambda_1, \lambda_2, \ldots, \lambda_n)$ represents the eigenvalues of the covariance matrix of the adversarial set, and $\|\boldsymbol{\lambda}\|_1$ is the sum of all eigenvalues, ensuring a normalized representation. The entropy function $H(\cdot)$ measures how evenly variance is distributed across eigenvectors, and $D$ denotes the dimensionality of the feature space, normalizing the complexity measure to the range $[0, 1]$. A higher complexity indicates a more uneven decision boundary, implying greater sensitivity to perturbations and risk of over-fitting. On the other hand, a lower complexity score suggests a smoother boundary, which is more likely to generalize well to unseen data.

## 5 Experimental Setup

In addition to reproducing the results from Wang & Shen (2024), in order to confirm the validity and robustness of the described methods, we aim to convey an empirically-driven intuition behind the GNNBoundary and to make it more accessible to practitioners and researchers. To reproduce the results, we utilized the authors' publicly available code repository[1]. However, several critical components were absent in the original implementation, as detailed in Appendix I. We implement the missing parts and make it public as linked in the abstract.

### 5.1 Adjacency Discovery

To verify claim 1, we run the algorithm for identifying adjacent class pairs proposed by Wang & Shen for all datasets. To further verify the resulting adjacency scores, we analyze the measured success rates [2] for near-boundary graph generation on the adjacent class pairs as well as on a selection of non-adjacent class pairs. High success rates would generally be expected on adjacent class pairs and low success rates would be expected on non-adjacent class pairs.

---

[1]The authors' GitHub repository can be accessed at: https://github.com/yolandalalala/GNNBoundary
[2]The success rate denotes the proportion of 1,000 runs that successfully generated a boundary graph within 500 iterations and within the target range of $[0.45, 0.55]$.

## 5.2 Datasets

The GNNBoundary method was originally evaluated on three datasets: Motif, Collab, and Enzymes, with the latter two being real-world graph datasets. The synthetic Motif dataset, introduced in Wang & Shen (2023), consists of graphs labeled with one of four predefined motifs: House, HouseX, Comp4, and Comp5. The Collab dataset comprises ego-networks of scientific collaboration, representing co-authorship networks in the fields of High Energy Physics (HE), Condensed Matter Physics (CM), and Astrophysics (Astro) Yanardag & Vishwanathan (2015). The Enzymes dataset Schomburg et al. (2004) consists tertiary-structured proteins classified into one of six enzyme classes.

Given that only 3 datasets are used in Wang & Shen (2024), we further assess the method's performance in real-world settings, using two additional real-world datasets: IMDB and Reddit-Mulitclass. The IMDB dataset consists of 1,500 graphs, where nodes represent actors and actresses. An edge is formed between two nodes if the corresponding individuals appeared in the same movie. Each graph is categorized into one of three genres: Comedy, Romance, or Sci-Fi. The Reddit dataset contains 232,965 posts collected from the social media platform Reddit. Each graph instance represents a subreddit community, where nodes correspond to posts, and edges indicate that a user commented on both posts. The dataset includes five class labels corresponding to different subreddits: *worldnews*, *videos* (general video-sharing), *AdviceAnimals* (humorous advice posts featuring animals), *Aww* (cute content), and *Mildlyinteresting*. This dataset contains larger graphs compared to the other datasets (cf. table 5) to test the scalability of the method.

## 5.3 Hyperparameters

Wang & Shen report one set of hyperparameters for all datasets. Moreover, it is missing configurations for the dynamic regularization scheduler as well as the graph target size. Consequently, we ran Bayesian Optimization Shahriari et al. (2015) to find the optimal hyperparameter configuration for each dataset. The search space consists of the sample size $K$, the target size, the target probabilities, the learning rate, the temperature, and the weight budget increase for the dynamic regularization scheduler and the weight budget decrease. Details on the search space are given in appendix H. Moreover, we employed a simple custom loss for the hyperparameter tuning, being the average deviation of the class probabilities from the target 0.5:

$$\mathcal{L}(CP) = \frac{1}{N_{cp}} \sum_{cp} \frac{|p(c_1^{cp}) - 0.5| + |p(c_2^{cp}) - 0.5|}{2}$$

where $N_{cp}$ is the number of class pairs in adjacent class-pair set $CP$ and $c_1^{cp}$ & $c_2^{cp}$ are class one and two of a class pair respectively. We chose this simple criterion instead of GNNBoundaries' dynamic boundary criterion to be independent of the tested method as well as for simplicity and thus easier interpretability.

## 5.4 Baselines

Wang & Shen propose a random baseline that generates boundary graphs using a graph from $c_1$ and a graph from $c_2$ and combines them with a random edge.

**Dataset-based Graph Sampling.** In the absence of prior methods for analyzing the decision boundary of GNN-based classifiers, the authors proposed a simple baseline that samples a graph from the dataset for each class in the class pair and connects them with a randomly assigned edge to obtain a graph that is theoretically representative of both classes. Thus, it is expected to activate the classifier's probability distributions maximally for these two classes. Our reproduction of this baseline corroborates the authors' findings, demonstrating that this approach frequently fails to produce near-boundary graphs, as the randomly sampled graph structures often lack the properties needed to elicit near-boundary probability activations.

**GNNInterpreter-based Graph Sampling.** To address the limitations of the random sampling baseline, we devised a novel baseline leveraging GNNInterpreter (Wang & Shen, 2023). Specifically, we generate class-representative graphs using GNNInterpreter and connect them with a random edge. Since GNNInterpreter optimizes these graphs to maximize their respective class activations—unlike dataset-sampled graphs,

which may not lie in the decision region—we hypothesize that combining two such graphs preserves and integrates class-specific features. Through GNN message passing, this aggregation is expected to enhance the activation of both classes more effectively than dataset-sampled graphs, as it reinforces decision-relevant features optimized for class activation. One limitation is that it introduces significant complexity due to the hyperparameter tuning required for GNNInterpreter (e.g., generating hundreds of graphs per class). These challenges render the GNNInterpreter-based baseline infeasible in settings where finding class-representative graphs (e.g. using GNNInterpreter) is not needed anyways.

## 5.5 Boundary Embedding Discovery

The stochastic nature of GNNBoundary's graph generation process raises the question of how the distribution of these generated graphs is structured in embedding space (i.e., after applying the graph embedding function $\eta_{L-1}(G)$). A robust generation process should produce embeddings that are broadly distributed yet concentrated around meaningful decision boundaries, ensuring that generated graphs effectively capture diverse but class-relevant features. To challenge the boundary embeddings proposed by the authors, we implement a method that randomly initializes an embedding and iteratively optimizes it toward a point in embedding space where the discriminator assigns a probability within the target range of $[0.45, 0.55]$ for the corresponding class pairs. This approach allows us to probe the structure of the decision boundary and assess whether GNNBoundary embeddings naturally align with it or require extensive optimization. The primary goal is to analyze how the distribution of embeddings would appear if directly optimized in embedding space, particularly after visualizing its 2D representation using UMAP (McInnes et al., 2018) and PCA (Jolliffe, 2002). Formally, with $\eta_L$ being the last layer of discriminator function $f$ that assigns a probability to a given embedding, let $\mathbf{z}$ be an embedding, which does not necessarily have to correspond to an actual valid graph. We seek a near-boundary embedding $\mathbf{z}^*$ that satisfies $\mathbf{z}^* = \arg\min_{\mathbf{z}} \mathcal{L}_{CE}(\eta_L(\mathbf{z}), \mathbf{y}^*)$ where $\mathcal{L}_{CE}$ is the Cross-Entropy loss and $\mathbf{y}^*$ is the target vector of shape $C$. Suppose, we want to find a boundary embedding for class $i$ and $j$, then $y_i^* = y_j^* = 0.5$ and $y_k^* = 0 \ \forall k \neq i, j$. The optimization is performed over an initially random embedding $\mathbf{z}$ to cover most near-boundary regions in the embedding space. Note that this is only to find near-boundary points in the embedding space for the purpose of visualization and hence there is no need for $\mathbf{z}^*$ to be the embedding of a valid graph.

## 5.6 Boundary Statistics under Different Target Probability Ranges

The accuracy of boundary metric approximations depends on both the quantity and quality of near-boundary graphs. The quality refers to how close these graphs are to the boundary (defined as having a probability of 0.5 for $c_1$ and $c_2$). To investigate this, we evaluate the boundary statistics (complexity, margin and thickness) for different target probability ranges. We analyze how these statistics deviate from the proposed target range of $[0.45, 0.55]$, providing insight into the consistency and reliability of the suggested range. If the statistics remain stable across small shifts in the target range, it suggests that $[0.45, 0.55]$ is a reliable approximation of the actual decision boundary. In contrast, large variations in these statistics when adjusting the target range would indicate that our chosen range may poorly represent the true boundary, highlighting a potential need for adjustment.

# 6 Results & Discussion

To verify the stated claims 1, 2 and 3 we reproduce all experiments from Wang & Shen (2024) as well as the extensions described in section 5, report the results and discuss them in this section.

## 6.1 Reproduction of Results

Before contributing additional insights on the optimized hyperparameters, new real-world datasets, the new baseline and the boundary statistics, we attempt to reproduce the results from Wang & Shen (2024) as good as possible under their reported configurations (refer to table 8 in appendix H for an overview of the authors' hyperparameters). Since some hyperparameters are not explicitly stated in Wang & Shen (2024), we infer

the values and ranges from the demo notebooks provided in the authors' code base: initial weight budget as 1, the weight increase as 1.15, the decrease as 0.95, and the target size as 30.

**Boundary Graph Generation.** When comparing our table 1 with table 1 in Wang & Shen (2024), we observe that we were mostly able to obtain very similar results when it comes to the ability of GNNBoundary to generate near-boundary graphs with both class-probabilities being close to 0.5. An exception of this are the first four class-pairs of the Enzymes dataset which did not converge under the target probability range of $p_{min} = 0.45$ and $p_{max} = 0.55$. We hypothesize this discrepancy between our and the authors' results to be caused by subpar hyperparameter settings, slight differences in discriminator performance (cf. appendix E and figure 3 in Wang & Shen (2024)) and to some extend random variation. In terms of complexity, we mostly observe higher values, i.e. greater sensitivity to perturbations and overfitting, thus worse generalization to unseen graphs. As we reproduce their random baseline, we observe different values, but a similarly strong discrepancy between the class-probabilities and the targeted 0.5 probability as well as high standard deviations. Despite the small discrepancy regarding the Enzymes dataset, we can mostly confirm the results from Wang & Shen regarding the class-pair probabilities of GNNBoundary boundary graphs. We confirm the superiority of GNNBoundary over the random baseline without limitations.
Our proposed baseline, which leverages GNNInterpreter-based graph sampling, did not outperform the random baseline suggested by Wang & Shen (2024). The resulting class probabilities for our baseline are reported in Table 4. Along with the challenging hyperparameter tuning required for GNNInterpreter, these results highlight the need for more effective baselines for generating near-boundary graphs.

**Finding Adjacent Class Pairs.** We reproduce the results for the Collab, Motif and Enzymes datasets using their provided code and observe insignificant differences in adjacency scores (appendix B). As layed out in section 5.1, we also analyze the correlation between the adjacency score and the corresponding GNNBoundary success rate to verify the author's proposed adjacency discovery algorithm. Figure 1 depicts the measurements of the adjacency score and the GNNBoundary success rate for class pairs across datasets. Class pairs that were determined to be adjacent by Wang & Shen are depicted in blue, while a selection of non-adjacent class pairs is plotted in orange. The adjacency score used for fig. 1 is the one we obtained from the author's adjacency discovery algorithm.

As expected, we can see a clear correlation between the adjacency score and success rate, indicating that the obtained adjacency score is a good proxy for adjacency. To further quantify these results, we calculate the Pearson correlation coefficient and deter-

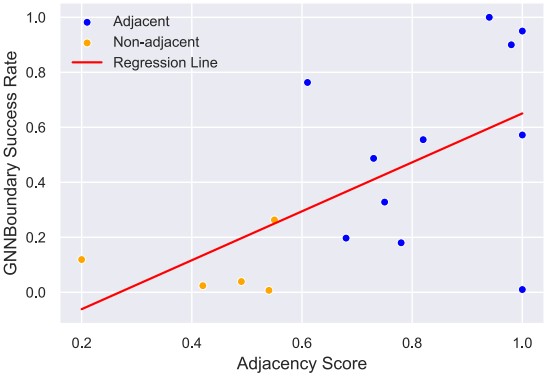

Figure 1: Comparison of success rate to adjacency score for all adjacent class pairs obtained using their adjacency finding algorithm (Collab, Motif and Enzymes datasets). Success rates are measured under optimized hyperparameters we obtained from HPO (cf. section 6.2). Adjacent means that the GNNBoundary author's reported these as adjacent.

mine if the slope coefficient is statistically significant under a 95% confidence level. The Pearson correlation coefficient is 0.6 and the $p$-value is 0.0139. Hence, there is a clear positive correlation and the regression is statistically significant under set confidence level.

*Given the clear positive correlation and statistical significance, we accept claim 1.*

**Success Rate in Boundary Graph Generation.** Under the authors' hyperparameter settings we cannot confirm the high reported success rates from Wang & Shen (2024) as well as we cannot confirm GNNBoundary consistently outperforming the cross-entropy baseline (cf. table 2) under their hyperparameter settings. When using our hyperparameter settings, we obtain success rates much closer to the authors' ones and consistent superiority over cross-entropy for the Motif and Collab dataset (cf. table 3). For Enzymes we again struggle to confirm their results even with optimized hyperparameters.

Table 1: Comparison of GNNBoundary and baseline near-boundary graph probabilities across datasets. We report the mean class probability among 500 boundary graphs and the corresponding standard deviation. No results were obtained for the Reddit dataset and convergence could not be reached for the "Romance-Sci-Fi" class pair (view section 6.3 for more details). **Probabilities closer to 0.5 are better**.

| Dataset | $c_1$ | $c_2$ | GNNBoundary | | | Random Baseline | |
|---------|-------|-------|-------------|---------|---------|-----------------|---------|
| | | | Complexity | $p(c_1)$ | $p(c_2)$ | $p(c_1)$ | $p(c_2)$ |
| **Motif** | House | HouseX | 0.02 | $0.51 \pm 0.03$ | $0.49 \pm 0.03$ | $0.72 \pm 0.26$ | $0.06 \pm 0.15$ |
| | House | Comp4 | 0.07 | $0.51 \pm 0.03$ | $0.49 \pm 0.03$ | $0.59 \pm 0.31$ | $0.38 \pm 0.32$ |
| | HouseX | Comp5 | 0.23 | $0.51 \pm 0.06$ | $0.49 \pm 0.06$ | $0.89 \pm 0.26$ | $0.00 \pm 0.02$ |
| **Collab** | HE | CM | 0.31 | $0.48 \pm 0.02$ | $0.48 \pm 0.02$ | $0.31 \pm 0.44$ | $0.00 \pm 0.00$ |
| | HE | Astro | 0.25 | $0.50 \pm 0.03$ | $0.49 \pm 0.03$ | $0.05 \pm 0.20$ | $0.95 \pm 0.20$ |
| **Enzymes** | EC1 | EC4 | 0.20 | $0.44 \pm 0.03$ | $0.43 \pm 0.02$ | $0.19 \pm 0.32$ | $0.30 \pm 0.43$ |
| | EC1 | EC5 | 0.11 | $0.43 \pm 0.02$ | $0.51 \pm 0.01$ | $0.20 \pm 0.30$ | $0.20 \pm 0.33$ |
| | EC1 | EC6 | 0.21 | $0.44 \pm 0.02$ | $0.49 \pm 0.04$ | $0.23 \pm 0.32$ | $0.01 \pm 0.07$ |
| | EC2 | EC3 | 0.32 | $0.45 \pm 0.04$ | $0.46 \pm 0.05$ | $0.13 \pm 0.29$ | $0.37 \pm 0.41$ |
| | EC4 | EC5 | 0.13 | $0.49 \pm 0.04$ | $0.49 \pm 0.04$ | $0.30 \pm 0.43$ | $0.20 \pm 0.33$ |
| | EC4 | EC6 | 0.36 | $0.46 \pm 0.04$ | $0.45 \pm 0.04$ | $0.33 \pm 0.38$ | $0.05 \pm 0.17$ |
| **IMDB** | Comedy | Romance | 0.03 | $0.46 \pm 0.01$ | $0.46 \pm 0.01$ | $0.30 \pm 0.14$ | $0.11 \pm 0.20$ |
| | Comedy | Sci-Fi | 0.27 | $0.50 \pm 0.03$ | $0.50 \pm 0.03$ | $0.30 \pm 0.13$ | $0.62 \pm 0.16$ |
| | Romance | Sci-Fi | – | – | – | – | – |

*To summarize, we confirm the ability of GNNBoundary to find faithful near-boundary graphs and its superiority over the random baseline. However we cannot confirm their high convergence rates. Under this limitation, we still consider claim 2 to be valid.*

**Boundary Statistics Analysis.** Based on the case study provided in section 5.3 of Wang & Shen (2024) we implement the boundary margin, boundary thickness and boundary complexity metrics and reproduce the results for their three datasets and report them in appendix E. We generally observe significantly higher values for the boundary thickness, which corresponds to a comparatively high probability density of graphs on the boundary as opposed to the class regions. This can be interpreted as worse separability.
For Motif, the margins are comparable for about half of the class combinations and significantly higher for the remaining half. Given that higher margins indicate better class-separability and with that better robustness, our results for Motif are even better than the ones reported by Wang & Shen. For Collab, we observe mostly lower margins, i.e. worse separability and robustness when compared to the authors results and for Enzymes we also observe mostly mixed results compared to the authors reports, where some are higher and other are lower without a clear tendency. In general, there are significant discrepancies between our results and the author's results, which makes us question the reliability and precision of the employed boundary statistics. Based on our observations, we would rather see these statistics as an approximate low-precision indicator of the boundary structure. Beyond that, we hypothesize the worse separability exhibited by a lower margin and higher thickness to be connected to a slightly worse classification performance as compared to the authors (cf. the confusion matrix in appendix E to figure 3 in Wang & Shen (2024)). This is despite the fact that we used the authors' provided discriminator model checkpoints.

*In the light of these findings, we accept claim 3 only under the strong limitation that the boundary statistics cannot be considered reliable or precise given their significant discrepancies under small changes of discriminator accuracy.*

While we are mostly able to confirm the results from Wang & Shen (2024) on the Motif and Collab datasets, we face challenges in reproducing the results for the Enzymes dataset throughout all experiments. Wang & Shen also observe class pair probabilities closer to the edges of the [0.45, 0.55] range, lower success rates

and higher boundary thickness for the Enzymes dataset in comparison to the other datasets. However, we cannot reach their results, not even under optimized hyperparameters (cf. table 1, table 3, appendix E). We hypothesize this to be connected to the high complexity of the dataset given the higher number of classes and worse classifier performance. This highlights the methods dependency on classifier performance and its limitations when it comes to more complex datasets. Our attempts to improve the performance for the Enzymes classifier or to use a different model architecture to enhance classifier performance did not succeed.

### 6.2   Hyperparameters

Among the hyperparameters that were reported by the authors we tuned the target probability range, the learning rate and the temperature. As can be seen in table 8 in appendix H, the discovered optimal configurations are highly dependent on the dataset and mostly deviate significantly from the author's reported configuration. This highlights the need for dataset specific reporting of hyperparameters and offers an explanation for the large discrepancy in convergence between our table 2 and table 2 in Wang & Shen (2024). Notably, the optimal values for temperature and learning rate can mostly be found at the boundaries of our tuning ranges designed around the values provided by the authors, which highlights that the optimal configurations are far from the values reported by Wang & Shen. These results are expected given that different datasets come with different optimization landscapes that require appropriate hyperparameters. For instance, an optimization landscape with a higher number of local minima usually requires a higher learning rate to enable global convergence.

### 6.3   New Datasets

We find that GNNBoundary works mostly well for IMDB, while it is not possible to run it in a reasonable amount of time for the Reddit dataset.

**IMDB Dataset.**   We obtain fast and stable convergence for boundary graphs in class pairs "Comedy-Romance" and "Comedy-SciFi", while we could not achieve convergence for class pair "Romance-SciFi". The same goes for GNNInterpreter training: we obtain good results for classes "Comedy" and "SciFi", but not for "Romance". We observe that the boundary graphs for "Romance-SciFi" are very sparsely connected (cf. appendix F.1) and usually even contain several disconnected graphs. This is inherently problematic given that GNNs are based on message-passing and thus likely the reason for the described convergence issues.

**Reddit Dataset.**   For training the boundary graph sampler, we increased the maximum number of nodes from 25 [3] to 550, which is slightly above the average graph size of 508 nodes (cf. table 5) to make convergence more likely. However, such a large graph has up to $\frac{n(n-1)}{2}$ undirected edges, which for 550 nodes would be up to 150.975 edges. Hence, the time and memory complexity of GNNBoundary is $\mathcal{O}(n^2)$ w.r.t. to the number of nodes $n$, leading to training times of around 330 minutes for a single boundary graph sampler on an Apple M3 chip. HPO and generating 500 boundary graphs to compute the boundary statistics was hence not possible under the resource constraints of this work. Thus, we cannot report results for this dataset.

### 6.4   Random Baseline for Boundary Graph Class-Probabilities

Comparing table 1 and table 4, we cannot observe that the baseline based on dataset graphs has class probabilities closer to 0.5 than the one based on GNNInterpreter graphs: on average 35.7% for $p(c_1)$ and 23.1% for $p(c_2)$ when using dataset class graphs whereas the average probabilities for GNNInterpreter-based class graphs are 37.3% for $p(c_1)$ and 20.0% for $p(c_2)$. The variances of the class probabilities across the GNNInterpreter- and dataset-based baselines are also comparable. We also investigated using 2, 3 and 5 random edges but did not observe significant changes in the results and hence omit further details on this.

---

[3]We used a maximum number of nodes of 25 per graph for the other datasets.

### 6.5 Robustness of Boundary Detection

As described in section 5.5, we attempt to inspect the classifier embedding space by plotting the 2D UMAP and PCA of each adjacent class pair, the corresponding GNNBoundary near-boundary graphs and the near-boundary embeddings discovered by our cross-entropy based method described in section 5.5. The visual results of our analysis can be found in appendix J. In general, we found that GNNBoundary graphs form a larger structure throughout the embedding space, while our boundary embeddings are concentrated in one location (see herefore fig. 7 and fig. 8). This is due to the fact that our cross-entropy based method optimizes for probabilities under the use of a Softmax instead of the logits. Given that the logits are unbounded, our method always resorts to the same minimum for which the difference in magnitude of the logits enable the desired probability distribution. In contrast, GNNBoundary graph embeddings circumvent this limitation, spreading across a broader structure in latent space, demonstrating its robustness and ability to generate a diverse range of meaningful boundary graphs. Given that direct optimization in embedding space did not yield high variability, we emphasize the significant achievement of GNNBoundary in generating graphs that effectively traverse constrained regions without requiring direct optimization in embedding space.

We note that PCA is not locality-preserving. Hence, the fact that the boundary graphs do not appear in-between the classes on the plot does not necessarily correspond to the actual proximity in high-dimensional latent space. Consequently, this empirical study might give an intuition for the distribution of embeddings in latent space, but it cannot be used to argue about the precise location of the found boundary graphs in that latent space.

### 6.6 Boundary Statistics under Different Target Probability Ranges

To validate the $[0.45, 0.55]$ target range $[p_{\min}, p_{\max}]$, we analyzed boundary statistics (thickness, margin, complexity) across multiple ranges from $[0.43, 0.57]$ to $[0.495, 0.505]$ (appendix K). This assessment verifies whether the range proposed by Wang & Shen (2024) reliably captures near-boundary graph properties.

Overall, the $[0.45, 0.55]$ range offers a reasonable trade-off for boundary graph generation. While some deviations—especially for narrower ranges like $[0.495, 0.505]$—are more pronounced due to the stochastic nature of graph sampling, most statistics remain consistent around this interval. For example, boundary thickness for House-Comp4 in Motif is stable, whereas Collab shows expected variations near the boundary. Notably, boundary margin deviations in the Enzymes dataset are less stable, likely because Enzymes is generally harder to optimize. As a result, the generated boundary graphs in Enzymes tend to be more dispersed in embedding space, leading to fluctuating margin values across different target ranges. Hence, this variation is not a drawback of the chosen target range but rather of the general instability of the boundary graph generation for the Enzymes dataset (see table 1 for success rates of graph generation for Enzymes). Despite these fluctuations, the $[0.45, 0.55]$ range presents a solid compromise: it is wide enough to ensure a sufficient number of near-boundary graphs for analysis, while still preserving meaningful class distinctions. Therefore, we conclude that this range balances statistical robustness and practical feasibility in boundary graph generation.

## 7 Conclusion and Next Steps

GNNBoundary is a pioneering framework that enables finding near-boundary graphs in GNN classifiers that can be used to collect the boundary statistics margin, thickness and complexity for a quantitative assessment of a GNN classifiers robustness. It is the first work to offer explainability on the decision boundary of a GNN classifier. In this study, we reproduced the author's results and confirm the author's claims: (1) we observe similar results when using their adjacency discovery algorithm and report a high correlation between the obtained adjacency scores and near-boundary graph generation success rates. Further, (2) we find that GNNBoundary can consistently generate near-boundary graphs and is superior to the cross-entropy baseline. However, this was only possible after extensive hyperparameter optimization. Finally, (3) we observe high variations on the boundary statistics, which makes us question their reliability and precision. Nonetheless, they can still provide some orientation on the generalization capabilities of the classifier. As a result, we confirm only limited applicability of the boundary statistics. Throughout experiments, we observe performance issues with the Enzymes dataset and explain this with it's relatively high complexity and low classifier performance. To further assess GNNBoundary's general applicability, we test the method on two

additional real-world datasets. In that, we observe fair results for one dataset (IMDB) and no results for the other (Reddit) due to a deficient scalability with larger graphs. Moreover, we conduct an empirical analysis of GNNBoundary's robustness in boundary detection and find it to be robust compared to a simple cross-entropy based baseline method for finding near-boundary embeddings that we introduce. Finally, we perform a study on the stability of the boundary statistics under different target class probability ranges and confirm the $[0.45, 0.55]$ range suggested by the authors.

In summary, we can commend the effort of the GNNBoundary authors as the interpretability of GNN classifiers is a valuable and underexplored topic, but issues around the usability of their method, including hurdles like the need for extensive hyperparameter optimization, confine future researchers ease to build upon this idea.

Future research could work on a more realistic graph sampling framework that overcomes the strong independence assumption being made by the independence of the distributions for edges, nodes and node features. Attempts to reduce the method's need for hyperparameter optimization or alternative approaches that are easier to apply could further contribute to GNN classifier explainability. Despite our extensive hyperparameter optimization efforts, we cannot confirm the author's consistently high convergence rates and observe significant discrepancies on the boundary statistics, highlighting the need for more robust methods.

## Acknowledgments

The authors would like to express their sincere gratitude to the lecturers and staff of the course *Fairness, Accountability, Confidentiality and Transparency in AI* at the University of Amsterdam for their insightful discussions and feedback. Additionally, we acknowledge Dr. Pascal Mettes (University of Amsterdam) for his guidance and explanations of topics covered in this research. Their insights and expertise contributed to our understanding and the development of this work.

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

# A  Related work

The explainability methods of GNNs review by Kakkad et al., as mentioned, defines two types of explainability approaches, a post-hoc and a self-interpretable approach. The main difference between these two approaches is *where* is the examination of the explanation taking place. In other words, post-hoc models examine the explainability of the model using only its inputs, outputs and at times the internal parameters of the model, while the self-interpretable methods design the explainability-architecture directly inside the model, such as a subgraph extraction module, that provides insights into the decision-making process while it is still on going. Self-interpretable methods are inherently more complex to conduct, as they would require access to the architecture of model being examined. Even when building a model with the intention of self-interpretability, the use of this method often sacrifices prediction accuracy for self-interpretability.

Post-hoc methods can be distinguished into white box methods, methods that require access to the model's parameters and embeddings (such as node weights of different layers), and black box methods, that do not require access to the embedding and the parameters and can, for example, train a surrogate model.

Such surrogate models, motivated by the complexity of the original models that might not be interpretable, try to approximate the relationship between input and output by simpler and interpretable functions of the surrogate model. A study by Huang et al. (2022), attempting an explainability method under the name GraphLime uses local explanations that rely on the Hilbert-Schmidt Independence Criterion Lasso (HSIC Lasso) model, a kernel-based, nonlinear feature selection technique designed for interpretability. This allows the method to operate under the assumption that the original node features in the graph are inherently interpretable. Given a node and its N-hop neighborhood, the HSIC Lasso method identifies a subset of node features that have the greatest influence on the model's prediction, which will then serve as the explanation for that prediction. On the other hand, the surrogate model can be a GNN as well, such as in the study by Pereira et al. (2023), the DistilnExplain model creates a simpler version of the original GNN model using knowledge distillation and then generates explanations by solving a basic convex optimization problem. It uses a lightweight surrogate model called Simplified Graph Convolution (SGC), which is a linear model without any nonlinear activation functions and uses just one weight matrix across all layers. The SGC model is trained to mimic the original GNN by minimizing the KL-divergence between their predictions. Once trained, explanations are obtained from the SGC model using a straightforward convex program.

The first white box method we investigate is the decomposition method, such as the DEGREE method by Feng et al. (2022). It finds an explanation in the form of subgraph structures. First, it decomposes the message-passing feed-forward propagation mechanism of the GNN to find a contribution score of a group of target nodes. Next, it uses an agglomeration algorithm that greedily finds the most influential subgraph as the explanation.

Further on, the important subgraphs used as explanations can be found by perturbing the input; these are known perturbation-based methods. GNNExplainer by Ying et al. (2019) is one of the earliest methods developed for explaining GNN predictions. It provides explanations by identifying a subgraph and a subset of node features that most strongly impact the model's output. To do this, it learns continuous masks over both the adjacency matrix and the node features, optimizing them to maximize the agreement between the model's prediction on the masked subgraph and the true class label using a cross-entropy loss.

Another approach, Zorro by Funke et al. (2022), provides explanations by identifying the most important nodes and features that maximize Fidelity. Fidelity is calculated as the expected validity of the perturbed input, and Zorro uses a greedy algorithm that selects the node and feature with the highest fidelity score at each step. Unlike other methods, Zorro applies a discrete mask to select a subgraph, without relying on backpropagation.

In contrast, ReFine by Wang et al. (2021) adopts a two-stage strategy involving edge attribution (pre-training) and edge selection (fine-tuning). During the pre-training phase, a GNN and an MLP are trained to estimate edge probabilities for the entire class by maximizing mutual information and applying a contrastive loss between classes. In the fine-tuning phase, these edge probabilities are used to sample edges and generate instance-specific explanations by optimizing mutual information for that particular prediction. While decomposition and perturbation methods focus on analyzing and manipulating existing components

of the input graph or the model's computations to derive explanations, generation-based approaches take a different path. Instead of selecting or masking parts of the input, these methods aim to directly generate new graph structures or features that are inherently interpretable. By framing explanation as a generative task, these methods provide more flexible and often more human-graspable insights, which highlights the need for generation-based methods of explainability. The method used as a baseline in both the initial GNNInterpreter paper and the reproduction paper of it was the XGNN method by Yuan et al. (2020). This method offers model-level explanations by generating representative subgraph patterns that maximize the prediction confidence for a specific class. The subgraph is constructed using a reinforcement learning-based graph generator, which is trained through policy gradient optimization. In this framework, the current graph represents the state, adding an edge corresponds to an action, and the reward is derived from the model's prediction score combined with predefined constraints. These rules are typically grounded in domain knowledge, ensuring that the generated subgraphs are both meaningful and plausible. In contrast to the XGNN method comes the GNNInterpreter (Wang & Shen, 2023), a method developed by the authors of GNNBoundary. GNNInterpreter is a method that aims at generating the explanation (interpreter) graphs for each class. The similarity between the explanation graph embedding and the mean embedding of all graphs act as an optimization constraint, while the objective is to maximize the likelihood of obtaining the explanation graph. It doesn't require domain knowledge as opposed to XGNN and the study claims it outperforms XGNN, the state of the art model at the time, in generating explanation graphs aligned better with the dataset. Interestingly, the RE:GNNInterpreter study by Vasilcoiu et al. tried to replicate these results and failed, concluding it in fact doesn't outperform XGNN.

This leads us to the follow-up study to GNNInterpreter, which builds upon a similar core concept. GNNBoundary by Wang & Shen (2024) adopts a generation-based approach to enhance interpretability, offering a promising direction for explainable GNNs. It aims to build upon the original GNNInterpreter architecture, which previously faced reproducibility challenges. Given the method's potential importance an impact on the explainability of GNNs, it is essential to investigate into its reproducibility and reliability.

## B   Adjacency Finding

Adjacency rates reported in the original work differ slightly from the values obtained when reproducing adjacency tables, using the checkpoints provided and the seed noted in the open source code. As the differences are minimal, we decide upon accepting the adjacent pairs stated by the authors and proceed further with the research using them.

Figure 2: Adjacency matrices for each dataset. Small adjacency values indicate that no decision boundary between the two classes is present in the classifier. A threshold of 0.8 is applied to explore the *adjacent class-pairs*, while the class-pairs below this threshold are considered *non-adjacent*.

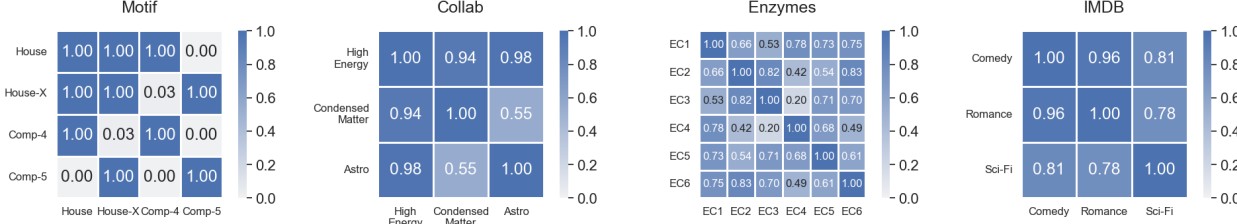

## C   Success Rates of Boundary Graph Generation

The success rate measures the proportion of 1000 runs in which the boundary graph generation converged (generated a boundary graph in the target range of $[0.45, 0.55]$) within 500 iterations. Given that the reported success rates were not achieved using the authors' implementation, HPO was used to find improved configurations. Given the discrepancy between the success rates using the author's hyperparameters and

our optimized ones with the reported results, we consider their report of experimental setup insufficient for reproducing their work.

Table 2: Convergence success rates **under the author's hyperparameters** across class pairs comparing GNNBoundary to the baseline cross-entropy criterion for success rate and average convergence iteration. **Higher success rate is better**.

| Dataset | $c_1$ | $c_2$ | Success Rate | | Avg. Convergence Iteration | |
| --- | --- | --- | --- | --- | --- | --- |
| | | | GNN-Boundary | Cross Entropy | GNN-Boundary | Cross Entropy |
| **Motif** | House | HouseX | **0.072** | 0.055 | **125.24** | 273.40 |
| | House | Comp4 | **0.681** | 0.653 | 79.86 | **64.59** |
| | HouseX | Comp5 | 0.000 | **0.002** | – | **429.50** |
| **Collab** | HE | CM | 0.793 | **0.998** | 118.00 | **65.30** |
| | HE | Astro | **0.998** | 0.923 | **12.38** | 109.27 |
| **Enzymes** | EC1 | EC4 | **0.120** | 0.004 | 216.62 | **103.25** |
| | EC1 | EC5 | 0.054 | **0.631** | 222.30 | **174.01** |
| | EC1 | EC6 | 0.184 | **0.853** | 241.34 | **51.63** |
| | EC2 | EC3 | 0.242 | **0.776** | 158.28 | **103.92** |
| | EC4 | EC5 | 0.102 | **0.720** | **103.84** | 157.04 |
| | EC5 | EC6 | 0.440 | **0.972** | **59.08** | 72.80 |

Table 3: Convergence success rates **under optimized hyperparameters** across class pairs comparing GNNBoundary to the baseline cross-entropy criterion for success rate and average convergence iteration. **Higher success rate is better**.

| Dataset | $c_1$ | $c_2$ | Success Rate | | Avg. Convergence Iteration | |
| --- | --- | --- | --- | --- | --- | --- |
| | | | GNN-Boundary | Cross Entropy | GNN-Boundary | Cross Entropy |
| **Motif** | House | HouseX | **0.527** | 0.055 | **201.65** | 273.40 |
| | House | Comp4 | **0.950** | 0.653 | **41.45** | 64.59 |
| | HouseX | Comp5 | **0.010** | 0.002 | **343.23** | 429.50 |
| **Collab** | HE | CM | **1.000** | 0.998 | 77.75 | **65.30** |
| | HE | Astro | **0.900** | 0.923 | **19.71** | 109.27 |
| **Enzymes** | EC1 | EC4 | **0.180** | 0.004 | 216.62 | **103.25** |
| | EC1 | EC5 | 0.487 | **0.631** | 222.30 | **174.01** |
| | EC1 | EC6 | 0.328 | **0.853** | 241.34 | **51.63** |
| | EC2 | EC3 | 0.555 | **0.776** | 158.28 | **103.92** |
| | EC4 | EC5 | 0.197 | **0.720** | **103.84** | 157.04 |
| | EC5 | EC6 | 0.763 | **0.972** | **59.08** | 72.80 |
| **IMDB** | Comedy | Romance | **0.750** | 0.0720 | **86.11** | 182.92 |
| | Comedy | Sci-Fi | **0.770** | 0.5080 | **228.52** | 243.43 |
| | Romance | Sci-Fi | – | – | – | – |

No results were obtained for the Reddit dataset (cf. section 6.3 for more details) and convergence could not be reached for the "Romance-SciFi" class pair (cf. section 6.3).

# D   Baseline using GNNInterpreter

Given some ambiguities in Wang & Shen (2024) and the fact that GNNInterpreter graphs are used for the boundary statistics, we also compute the random baseline using GNNInterpreter graphs. It works the same way as the standard random baseline with the only difference being that it samples class-graphs from previously trained GNNInterpreter samplers instead of the labeled training datasets themselves.

Table 4: Baseline probabilities for datasets and class tuples sampling using **GNNInterpreter class graphs** connected using a random edge to obtain boundary graphs.

| Dataset | Class Pairs | | Novel Baseline | |
|---|---|---|---|---|
| | $c_1$ | $c_2$ | $p(c_1)$ | $p(c_2)$ |
| **Motif** | House | HouseX | $0.3799 \pm 0.4140$ | $0.0742 \pm 0.2260$ |
| | House | Comp4 | $0.3171 \pm 0.3784$ | $0.4397 \pm 0.4307$ |
| | HouseX | Comp5 | $0.2394 \pm 0.3648$ | $0.4669 \pm 0.4934$ |
| **Collab** | HE | CM | $0.9607 \pm 0.0008$ | $0.0000 \pm 0.0000$ |
| | HE | Astro | $0.9607 \pm 0.0008$ | $0.0393 \pm 0.0008$ |
| **Enzymes** | EC1 | EC4 | $0.0360 \pm 0.1575$ | $0.0272 \pm 0.1551$ |
| | EC1 | EC5 | $0.0736 \pm 0.2442$ | $0.0250 \pm 0.1501$ |
| | EC1 | EC6 | $0.4461 \pm 0.4790$ | $0.0010 \pm 0.0160$ |
| | EC2 | EC3 | $0.0748 \pm 0.2462$ | $0.6005 \pm 0.4589$ |
| | EC4 | EC5 | $0.3377 \pm 0.4299$ | $0.5146 \pm 0.4564$ |
| | EC5 | EC6 | $0.2760 \pm 0.4256$ | $0.0211 \pm 0.1212$ |

# E   Boundary Statistics

We run the boundary complexity, margin and thickness as well as a confusion matrix for all three datasets from Wang & Shen (2024) and report on all of them except for the boundary complexity (which can be found in table 1) in the following fig. 3. These results are discussed in section 6.1.

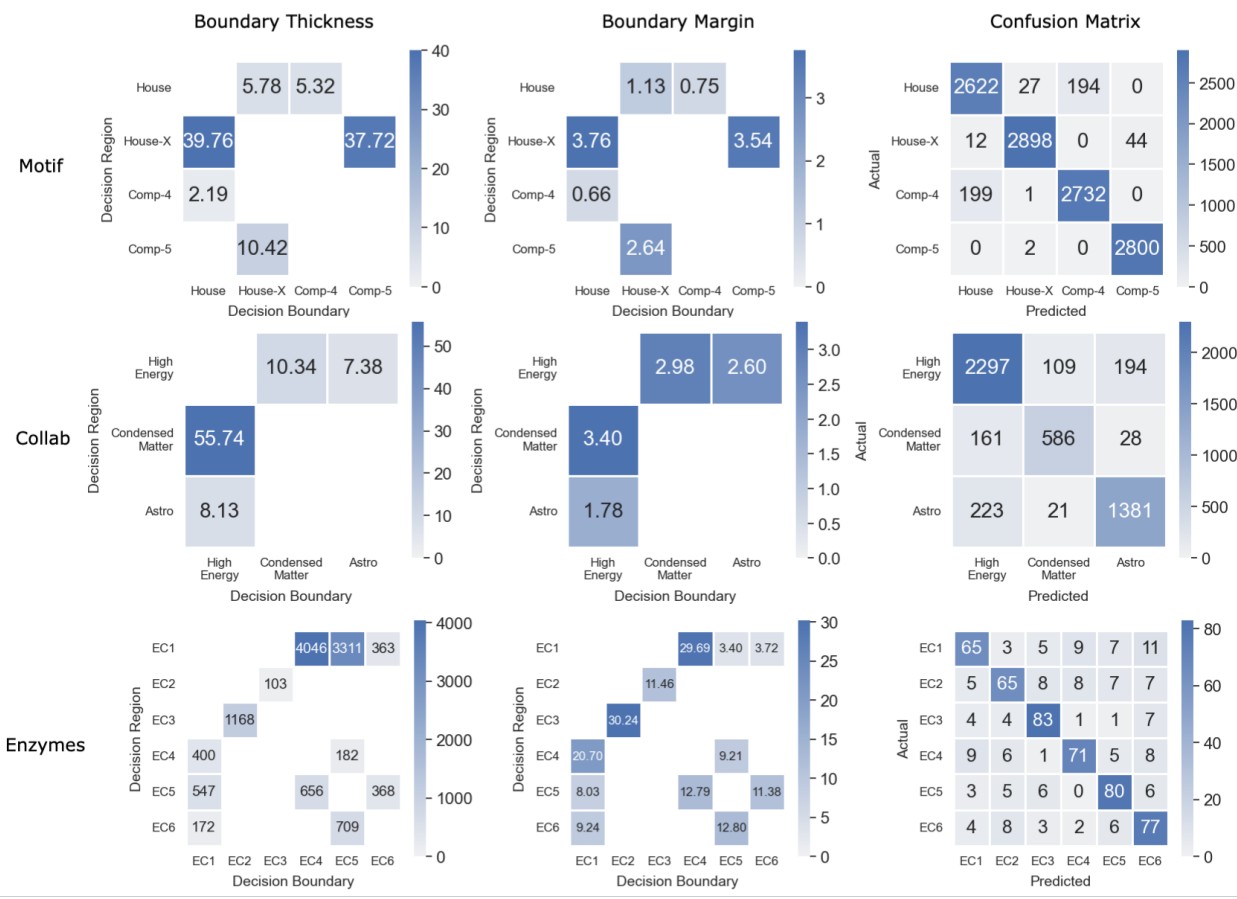

Figure 3: Boundary metric analyses and confusion matrices.

# F   Datasets

Apart from the datasets we added, Wang & Shen (2024) employ the Motif, Collab and Enzymes datasets described in the following. For a better understanding of the employed datasets, we provide table 5.

Table 5: Dataset statistics. Size refers to the number of nodes in a graph.

| Statistic | Size (Nodes) | Mean Graph Size | Median Graph Size | Min Graph Size | Max Graph Size | Std Graph Size |
|---|---|---|---|---|---|---|
| Motif | 11,531 | 57.07 | 51.00 | 14 | 112 | 25.65 |
| Collab | 5,000 | 74.49 | 52.00 | 32 | 492 | 62.30 |
| Enzymes | 600 | 32.46 | 32.00 | 2 | 125 | 14.87 |
| Reddit | 4,999 | 508.51 | 374.00 | 22 | 3,648 | 452.57 |
| IMDB | 1,500 | 13.0 | 10.0 | 7 | 89 | 8.52 |

### F.1 Details on the IMDB Dataset

Following the challenges on generating boundary graphs for the "Romance-SciFi" class pair and GNNInterpreter graphs for the "Romance" class, we provide examples of "Romance"-graphs and "Romance-SciFi"-graphs to identify potential root causes.

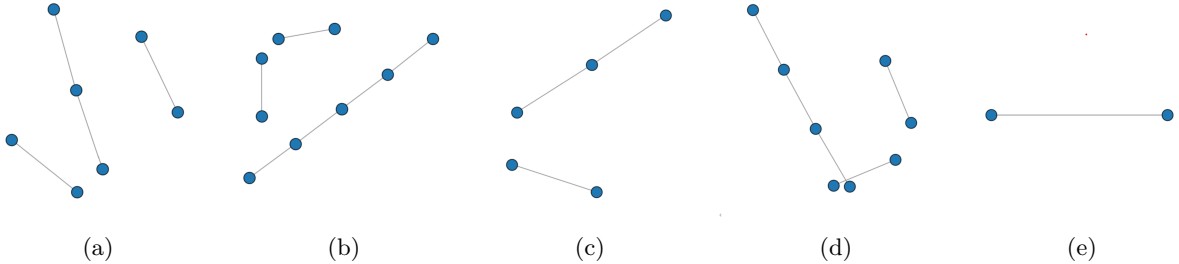

(a)        (b)        (c)        (d)        (e)

Figure 4: Examples of Romance-SciFi near-boundary graphs.

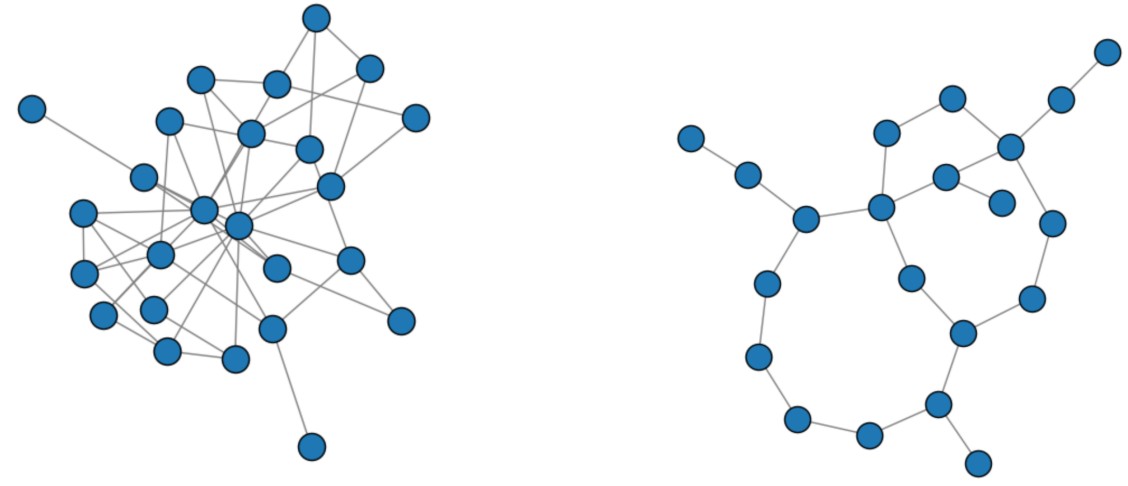

(a) Near-boundary graph for Romance-Comedy.        (b) Near-boundary graph for Comedy-SciFi.

Figure 5: Comparison of near-boundary graphs for Romance-Comedy and Comedy-SciFi class pairs.

## G   GCN Classifier

Table 6: Classifier accuracies, class-wise F1-scores, and GNN architecture hyperparameters for each dataset, using pre-trained checkpoints from the authors' work. For IMDB, the classifier was trained with architecture parameters informed by the dataset's graph properties.

| Dataset | Test Accuracy | Class | F1 Score | Architecture | |
|---|---|---|---|---|---|
| | | | | Hidden Channels | Num Layers |
| **Motif** | 0.961 | House | 0.923 | 6 | 3 |
| | | House X | 0.984 | | |
| | | Comp 4 | 0.946 | | |
| | | Comp 5 | 0.994 | | |
| **Collab** | 0.782 | High Energy | 0.802 | 64 | 5 |
| | | Condensed Matter | 0.617 | | |
| | | Astro | 0.823 | | |
| **Enzymes** | 0.483 | EC1 | 0.273 | 32 | 3 |
| | | EC2 | 0.526 | | |
| | | EC3 | 0.750 | | |
| | | EC4 | 0.348 | | |
| | | EC5 | 0.500 | | |
| | | EC6 | 0.500 | | |
| **IMDB** | 0.453 | Comedy | 0.156 | 64 | 5 |
| | | Romance | 0.491 | | |
| | | Sci-Fi | 0.573 | | |

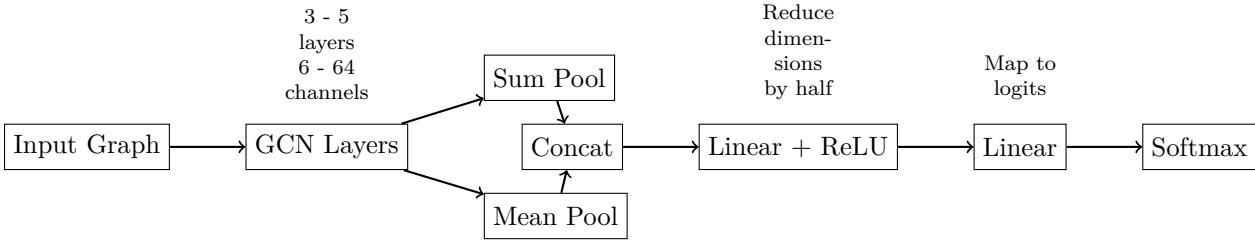

Figure 6: Architecture overview of the GCN classifier. The model processes input graphs through multiple GCN layers with LeakyReLU and Dropout, performs global weighted pooling operations, and uses linear layers for the final classification.

## H  Hyperparameter Optimization

The following table shows the search space employed for automatic hyperparameter optimization using Bayesian Optimization. We worked with 8 random starts, 200 tuning iterations and 3 runs per iteration. We found that the results within those 3 runs were mostly consistent and we took the mean performance across those 3 runs as the performance of the entire tuning iteration. Each run had 1000 training iterations. Hyperparameter optimization was done separately for each of the datasets.

Table 7: Search Space for Hyperparameters

| Hyperparameter | Data Type | Range / Values | Prior Distribution |
|---|---|---|---|
| Target Size | Integer | [20, 60] | Uniform |
| Target Probabilities | Categorical | {"0.45-0.55", "0.4-0.6", "0.35-0.65"} | Uniform |
| Learning Rate | Real | [0.01, 1] | Uniform |
| Temperature | Real | [0.05, 0.5] | Log-Uniform |
| Weight Budget inc. | Real | [1.05, 1.2] | Uniform |
| Weight Budget dec. | Real | [0.94, 0.99] | Uniform |

Table 8: Hyperparameter optimization results. The search space includes the sample size $K$, the target size, the target probabilities, the learning rate, the temperature, the weight budget increase for the dynamic regularization scheduler and the weight budget decrease. HPO for the Reddit dataset was not possible under the given configurations due to the high graph size (cf. section 6.3).

| Hyperparameter | Collab | Motif | Enzymes | IMDB | Reddit | Authors |
|---|---|---|---|---|---|---|
| Iterations | 1000 | 1000 | 1000 | 1000 | 1000 | 1000 |
| Sample Size $K$ | 32 | 32 | 32 | 32 | 32 | 32 |
| Init. Weight Budget | 1 | 1 | 1 | 1 | - | - |
| Target Size | 60 | 50 | 46 | 60 | - | - |
| Target Probabilities | [0.45, 0.55] | (0.4, 0.6) | (0.35, 0.65) | (0.4, 0.6) | - | [0.45, 0.55] |
| Learning Rate | 0.01 | 1.0 | 0.02 | 0.9 | - | 1.0 |
| Temperature | 0.05 | 0.05 | 0.5 | 0.49 | - | 0.15 |
| Weight Budget Inc. | 1.20 | 1.05 | 1.10 | 1.12 | - | - |
| Weight Budget Dec. | 0.99 | 0.99 | 0.95 | 0.94 | - | - |
| Custom Loss | 0.038 | 0.118 | 0.170 | 0.10 | - | - |

## I  Remarks on Implementation

We commend the authors for their efforts to ensure the reproducibility of their work by releasing their code, pre-trained model checkpoints, and providing detailed Readme instructions. In the spirit of transparency, we offer additional context on the implementation process and suggestions for improvement, with the hope of contributing to future learning opportunities in the field.

Reproducing the main results of the study required a significant investment of both time and computational resources. While the provided notebooks served as helpful examples, we encountered issues with hyper-parameter configurations as they led to convergence problems and were incomplete. Since no dedicated

training script was included to replicate the results in the original paper, we implemented our own training runs using the parameters specified in the paper or inferred from the notebooks. As shown in table 2, this configuration often failed to generate sufficient near-boundary graphs, making the reproduction of the results significantly harder. To address this, we conducted hyperparameter optimization (HPO) to derive class-pair-specific configurations that improved convergence. However, even with the optimized parameters, the success rates remained lower than the authors reported ones.

An implementation for the boundary complexity, thickness and margin was also a not included in the code publicized by the authors.

The environment setup was facilitated by both an environment.yml and a pyproject.toml file. The latter included a Git dependency from a common XAI library[4] used by both the GNNInterpreter (Wang & Shen (2023)) and GNNBoundary (Wang & Shen (2024)) repositories. This common library contains several crucial components, but its inheritance structure introduced implementation challenges. Specifically, overlapping versions of key components within the shared repository led to confusion. Notably, the absence of a clear training script, the lack of well-defined boundary metric calculations, and the presence of multiple versions of graph sampling implementations required significant effort to understand and resolve the intended design.

## J Embedding Space Inspection

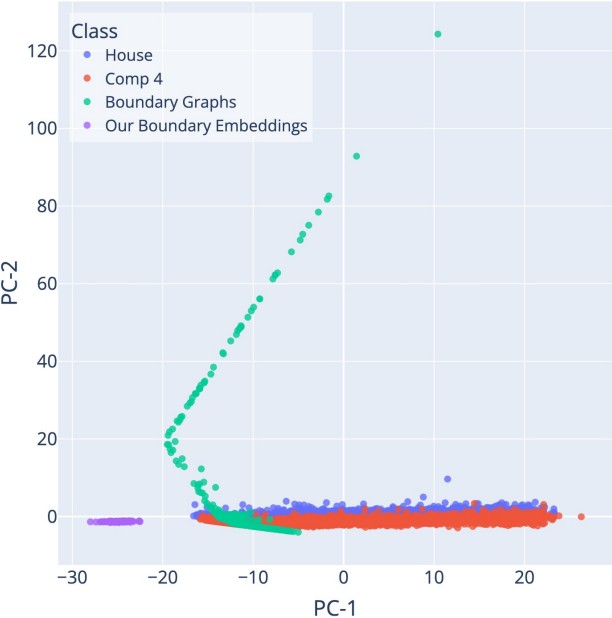

Figure 7: 2D PCA plot showing the first two principal components calculated from the classifier's embeddings of all graphs in the Motif class-pair House-Comp 4, the principal components of the corresponding GNNBoundary graph embeddings and our boundary embeddings.

---

[4]The gnn-common-xai repository can be found at: https://github.com/yolandalalala/gnn-xai-common

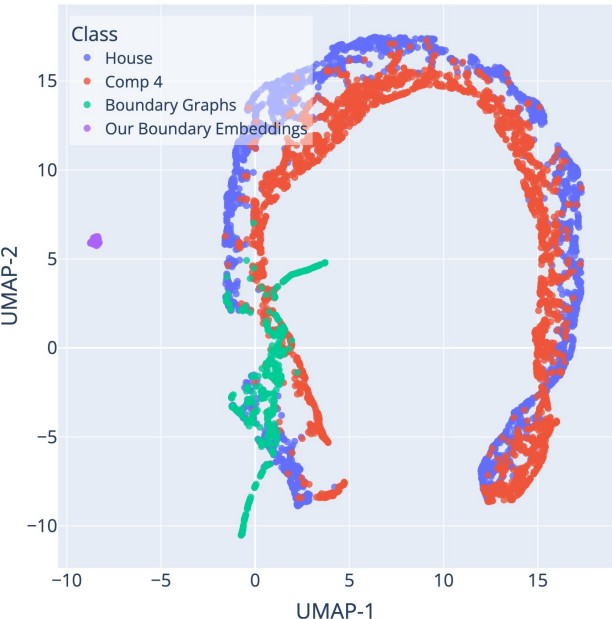

Figure 8: 2D UMAP plot for showing the classifier embeddings for the dataset, the GNNBoundary embeddings and our boundary embeddings for the Motif class pair House-Comp 4.

In fig. 8, we can see a very similar phenomenon as in fig. 7. The GNNBoundary near-boundary graph embeddings are spread across a larger area in a specific structure while the near-boundary embeddings are collapsed to the same location in the embedding space during optimization. This supports the findings that the GNNBoundary dynamic boundary criterion allows for a more robust optimization process than our cross-entropy based method. More details on our reasoning are provided in section 6.5.

# K   Relationship Between Boundary Metrics and Target Ranges

Table 9: Relationship between boundary thickness and target range

| Dataset | Class | (0.43,0.57) | (0.44,0.56) | (0.45,0.55) | (0.47,0.53) | (0.48,0.52) | (0.49,0.51) | (0.495,0.505) |
|---|---|---|---|---|---|---|---|---|
| **Motif** | | | | | | | | |
| | House | 21.860 | 21.860 | 21.860 | 21.039 | 20.657 | 22.296 | 24.124 |
| | Comp-4 | 23.006 | 23.006 | 23.006 | 22.959 | 22.366 | 23.980 | 24.727 |
| | HouseX | 94.832 | 94.108 | 92.528 | 89.067 | 87.809 | 95.975 | 95.114 |
| | Comp-5 | 49.071 | 48.010 | 46.773 | 46.545 | 46.309 | 48.020 | 49.969 |
| | House | 36.431 | 36.431 | 36.431 | 34.393 | 35.824 | 35.774 | 37.607 |
| | HouseX | 65.062 | 65.062 | 65.062 | 64.079 | 62.873 | 68.698 | 64.628 |
| **Collab** | | | | | | | | |
| | High Energy | 69.331 | 69.331 | 69.331 | 73.337 | 69.895 | 93.291 | 44.445 |
| | Astro | 2047.92 | 2047.92 | 2047.92 | 1605.01 | 1741.06 | 2198.34 | 3204.41 |
| | High Energy | 222.807 | 222.807 | 222.807 | 209.575 | 160.713 | - | - |
| | Condensed Matter | 36.111 | 36.111 | 36.111 | 25.734 | 22.746 | - | - |
| **Enzymes** | | | | | | | | |
| | EC4 | 205.928 | 206.793 | 210.382 | 220.714 | 219.387 | 230.614 | 229.475 |
| | EC5 | 286.960 | 282.358 | 292.259 | 308.021 | 300.307 | 363.206 | 298.949 |
| | EC5 | 207.833 | 209.161 | 204.640 | 180.781 | 162.449 | 163.469 | - |
| | EC6 | 115.654 | 110.911 | 114.174 | 75.252 | 77.022 | 84.510 | - |
| | EC1 | 272.131 | 272.131 | 303.868 | 327.851 | - | - | - |
| | EC5 | 255.304 | 255.304 | 333.342 | 272.701 | - | - | - |
| | EC1 | 374.599 | 315.357 | 143.406 | - | - | - | - |
| | EC4 | 174.381 | 105.329 | 75.279 | - | - | - | - |
| | EC1 | 216.613 | 209.391 | 227.606 | 151.933 | - | - | - |
| | EC6 | 100.109 | 77.762 | 74.587 | 48.068 | - | - | - |
| | EC2 | 91.520 | 96.992 | 86.903 | 77.186 | 81.010 | 113.270 | 98.641 |
| | EC3 | 1192.97 | 1192.71 | 1052.94 | 277.830 | 453.337 | 67.647 | 43.763 |

Table 10: Percentage deviation from (0.45,0.55) target range for boundary thickness

| Dataset | Class | (0.43,0.57) | (0.44,0.56) | (0.45,0.55) | (0.47,0.53) | (0.48,0.52) | (0.49,0.51) | (0.495,0.505) |
|---|---|---|---|---|---|---|---|---|
| **Motif** | | | | | | | | |
| | House | 0.00% | 0.00% | 0.00% | -3.76% | -5.50% | 1.99% | 10.36% |
| | Comp-4 | 0.00% | 0.00% | 0.00% | -0.20% | -2.78% | 4.23% | 7.48% |
| | HouseX | 2.49% | 1.71% | 0.00% | -3.74% | -5.10% | 3.73% | 2.79% |
| | Comp-5 | 4.91% | 2.64% | 0.00% | -0.49% | -0.99% | 2.67% | 6.83% |
| | House | 0.00% | 0.00% | 0.00% | -5.59% | -1.67% | -1.80% | 3.23% |
| | HouseX | 0.00% | 0.00% | 0.00% | -1.51% | -3.36% | 5.59% | -0.67% |
| **Collab** | | | | | | | | |
| | High Energy | 0.00% | 0.00% | 0.00% | 5.78% | 0.81% | 34.56% | -35.89% |
| | Astro | 0.00% | 0.00% | 0.00% | -21.63% | -14.98% | 7.35% | 56.47% |
| | High Energy | 0.00% | 0.00% | 0.00% | -5.94% | -27.87% | - | - |
| | Condensed Matter | 0.00% | 0.00% | 0.00% | -28.74% | -37.01% | - | - |
| **Enzymes** | | | | | | | | |
| | EC4 | -2.12% | -1.71% | 0.00% | 4.91% | 4.28% | 9.62% | 9.08% |
| | EC5 | -1.81% | -3.39% | 0.00% | 5.39% | 2.75% | 24.28% | 2.29% |
| | EC5 | 1.56% | 2.21% | 0.00% | -11.66% | -20.62% | -20.12% | - |
| | EC6 | 1.30% | -2.86% | 0.00% | -34.09% | -32.54% | -25.98% | - |
| | EC1 | -10.44% | -10.44% | 0.00% | 7.89% | - | - | - |
| | EC5 | -23.41% | -23.41% | 0.00% | -18.19% | - | - | - |
| | EC1 | 161.22% | 119.91% | 0.00% | - | - | - | - |
| | EC4 | 131.65% | 39.92% | 0.00% | - | - | - | - |
| | EC1 | -4.83% | -8.00% | 0.00% | -33.25% | - | - | - |
| | EC6 | 34.22% | 4.26% | 0.00% | -35.55% | - | - | - |
| | EC2 | 5.31% | 11.61% | 0.00% | -11.18% | -6.78% | 30.34% | 13.51% |
| | EC3 | 13.30% | 13.27% | 0.00% | -73.61% | -56.95% | -93.58% | -95.84% |

Table 11: Relationship between boundary margin and target range

| Dataset | Class | (0.43,0.57) | (0.44,0.56) | (0.45,0.55) | (0.47,0.53) | (0.48,0.52) | (0.49,0.51) | (0.495,0.505) |
|---|---|---|---|---|---|---|---|---|
| **Motif** | | | | | | | | |
| | House | 0.337 | 0.337 | 0.337 | 0.337 | 0.123 | 0.123 | 0.915 |
| | Comp-4 | 0.416 | 0.416 | 0.416 | 0.416 | 0.220 | 0.347 | 0.744 |
| | HouseX | 0.492 | 0.755 | 0.755 | 0.755 | 0.909 | 3.046 | 3.046 |
| | Comp-5 | 1.626 | 3.797 | 3.797 | 3.797 | 4.480 | 4.480 | 7.069 |
| | House | 1.236 | 1.236 | 1.236 | 1.503 | 1.430 | 1.430 | 1.922 |
| | HouseX | 0.972 | 0.972 | 0.972 | 0.972 | 1.456 | 2.011 | 2.996 |
| **Collab** | | | | | | | | |
| | High Energy | 3.023 | 3.023 | 3.023 | 2.598 | 3.333 | 3.318 | 4.097 |
| | Astro | 2.796 | 2.796 | 2.796 | 2.689 | 2.979 | 12.058 | 9.505 |
| | High Energy | 1.997 | 1.997 | 1.997 | 1.997 | 2.497 | - | - |
| | Condensed Matter | 1.645 | 1.645 | 1.645 | 1.645 | 1.982 | - | - |
| **Enzymes** | | | | | | | | |
| | EC4 | 17.748 | 17.748 | 17.748 | 32.503 | 32.503 | 32.300 | 34.783 |
| | EC5 | 4.815 | 4.815 | 4.815 | 33.999 | 33.999 | 33.633 | 33.633 |
| | EC5 | 3.045 | 3.045 | 3.045 | 17.337 | 24.270 | 16.537 | - |
| | EC6 | 11.425 | 11.425 | 11.425 | 10.411 | 14.968 | 6.294 | - |
| | EC1 | 17.251 | 17.251 | 18.542 | 18.542 | - | - | - |
| | EC5 | 12.642 | 12.642 | 24.377 | 24.377 | - | - | - |
| | EC1 | 5.919 | 4.399 | 28.629 | - | - | - | - |
| | EC4 | 13.383 | 12.763 | 28.059 | - | - | - | - |
| | EC1 | 20.664 | 20.677 | 9.465 | 25.900 | - | - | - |
| | EC6 | 13.949 | 15.315 | 10.926 | 36.139 | - | - | - |
| | EC2 | 12.138 | 3.154 | 3.154 | 18.348 | 18.348 | 27.913 | 37.612 |
| | EC3 | 9.944 | 10.469 | 14.045 | 18.061 | 18.061 | 27.764 | 34.422 |

Table 12: Percentage deviation from (0.45,0.55) target range for boundary margin

| Dataset | Class | (0.43,0.57) | (0.44,0.56) | (0.45,0.55) | (0.47,0.53) | (0.48,0.52) | (0.49,0.51) | (0.495,0.505) |
|---|---|---|---|---|---|---|---|---|
| **Motif** | | | | | | | | |
| | House | 0.00% | 0.00% | 0.00% | 0.00% | -63.50% | -63.50% | 171.51% |
| | Comp-4 | 0.00% | 0.00% | 0.00% | 0.00% | -47.12% | -16.59% | 78.85% |
| | HouseX | -34.83% | 0.00% | 0.00% | 0.00% | 20.40% | 303.44% | 303.44% |
| | Comp-5 | -57.18% | 0.00% | 0.00% | 0.00% | 17.99% | 17.99% | 86.17% |
| | House | 0.00% | 0.00% | 0.00% | 21.60% | 15.70% | 15.70% | 55.50% |
| | HouseX | 0.00% | 0.00% | 0.00% | 0.00% | 49.79% | 106.89% | 208.23% |
| **Collab** | | | | | | | | |
| | High Energy | 0.00% | 0.00% | 0.00% | -14.06% | 10.25% | 9.76% | 35.53% |
| | Astro | 0.00% | 0.00% | 0.00% | -3.83% | 6.55% | 331.26% | 239.95% |
| | High Energy | 0.00% | 0.00% | 0.00% | 0.00% | 25.04% | - | - |
| | Condensed Matter | 0.00% | 0.00% | 0.00% | 0.00% | 20.49% | - | - |
| **Enzymes** | | | | | | | | |
| | EC4 | 0.00% | 0.00% | 0.00% | 83.14% | 83.14% | 81.99% | 95.98% |
| | EC5 | 0.00% | 0.00% | 0.00% | 606.11% | 606.11% | 598.50% | 598.50% |
| | EC5 | 0.00% | 0.00% | 0.00% | 469.36% | 697.04% | 443.09% | - |
| | EC6 | 0.00% | 0.00% | 0.00% | -8.88% | 31.01% | -44.91% | - |
| | EC1 | -6.96% | -6.96% | 0.00% | 0.00% | - | - | - |
| | EC5 | -48.14% | -48.14% | 0.00% | 0.00% | - | - | - |
| | EC1 | -79.33% | -84.63% | 0.00% | - | - | - | - |
| | EC4 | -52.30% | -54.51% | 0.00% | - | - | - | - |
| | EC1 | 118.32% | 118.46% | 0.00% | 173.64% | - | - | - |
| | EC6 | 27.67% | 40.17% | 0.00% | 230.76% | - | - | - |
| | EC2 | 284.84% | 0.00% | 0.00% | 481.74% | 481.74% | 785.00% | 1092.52% |
| | EC3 | -29.20% | -25.46% | 0.00% | 28.59% | 28.59% | 97.68% | 145.08% |

Table 13: Relationship between boundary complexity and target range

| Dataset | Class | (0.43,0.57) | (0.44,0.56) | (0.45,0.55) | (0.47,0.53) | (0.48,0.52) | (0.49,0.51) | (0.495,0.505) |
|---|---|---|---|---|---|---|---|---|
| **Motif** | | | | | | | | |
| | House and Comp4 | 0.072 | 0.072 | 0.072 | 0.064 | 0.062 | 0.064 | 0.060 |
| | HouseX and Comp-5 | 0.182 | 0.135 | 0.114 | 0.080 | 0.057 | 0.058 | 0.066 |
| | House and HouseX | 0.015 | 0.015 | 0.015 | 0.012 | 0.010 | 0.008 | 0.007 |
| **Collab** | | | | | | | | |
| | High Energy and Astro | 0.253 | 0.253 | 0.253 | 0.229 | 0.217 | 0.240 | 0.253 |
| | High Energy and Condensed Matter | 0.313 | 0.313 | 0.313 | 0.298 | 0.343 | - | - |
| **Enzymes** | | | | | | | | |
| | EC4 and EC5 | 0.138 | 0.168 | 0.166 | 0.187 | 0.189 | 0.182 | 0.247 |
| | EC5 and EC6 | 0.349 | 0.326 | 0.335 | 0.403 | 0.413 | 0.374 | - |
| | EC1 and EC5 | 0.0455 | 0.0455 | 0.021 | 0.000 | - | - | - |
| | EC1 and EC4 | 0.199 | 0.122 | 0.189 | - | - | - | - |
| | EC1 and EC6 | 0.168 | 0.171 | 0.167 | 0.099 | - | - | - |
| | EC2 and EC3 | 0.306 | 0.289 | 0.296 | 0.313 | 0.348 | 0.596 | 0.000 |

Table 14: Percentage deviation from (0.45,0.55) target range for boundary complexity

| Dataset | Class | (0.43,0.57) | (0.44,0.56) | (0.45,0.55) | (0.47,0.53) | (0.48,0.52) | (0.49,0.51) | (0.495,0.505) |
|---|---|---|---|---|---|---|---|---|
| **Motif** | | | | | | | | |
| | House and Comp4 | 0.00% | 0.00% | 0.00% | -11.11% | -13.89% | -11.11% | -16.67% |
| | HouseX and Comp-5 | 59.65% | 18.42% | 0.00% | -29.82% | -50.00% | -49.12% | -42.11% |
| | House and HouseX | 0.00% | 0.00% | 0.00% | -20.00% | -33.33% | -46.67% | -53.33% |
| **Collab** | | | | | | | | |
| | High Energy and Astro | 0.00% | 0.00% | 0.00% | -9.49% | -14.23% | -5.14% | 0.00% |
| | High Energy and Condensed Matter | 0.00% | 0.00% | 0.00% | -4.79% | 9.58% | - | - |
| **Enzymes** | | | | | | | | |
| | EC4 and EC5 | -16.87% | 1.20% | 0.00% | 12.65% | 13.86% | 9.64% | 48.80% |
| | EC5 and EC6 | 4.18% | -2.69% | 0.00% | 20.30% | 23.28% | 11.64% | - |
| | EC1 and EC5 | 116.67% | 116.67% | 0.00% | -100.00% | - | - | - |
| | EC1 and EC4 | 5.29% | -35.45% | 0.00% | - | - | - | - |
| | EC1 and EC6 | 0.60% | 2.40% | 0.00% | -40.72% | - | - | - |
| | EC2 and EC3 | 3.38% | -2.36% | 0.00% | 5.74% | 17.57% | 101.35% | -100.00% |

