# OpenReview forum: "[RE] GNNBoundary: Finding Boundaries and Going Beyond Them"
_TMLR — Accepted by TMLR_

### Review · Reviewer_mFFx · 2025-03-04

**Summary Of Contributions:**

This work reevaluates and reproduces a previous ICLR publication, "GNNBoundary: Towards Explaining Graph Neural Networks through the Lens of Decision Boundaries." In addition, it further evaluates the previously proposed method on two additional datasets, IMDB and Reddit, where Reddit is not computable in all experiments. Lastly, the authors include some additional disparate evaluations. The authors confirm, for the most part, the claims made in the original work.

**Audience:**

Yes

**Claims And Evidence:**

Yes

**Requested Changes:**

- Please include a detailed related work section, paying special care to any work published since 2023. If this cannot fit into the main body of this work, include it as an appendix. Feel free to refer to the related work of the original work where appropriate. (acceptance)
- "Section 3.3 Decision region and boundary" does not fully cover the background of this work. namely, \mathcal B^{(l)}_{c_1||c_2} is not explained. What is this quantity, and what is it used for in this work? Please make sure that no reader needs to refer back to the original work of GNNBoundary. (acceptance)
- "Section 4 Notation paragraph" The first sentence is malformed: "A graph, ...." This sentence is missing multiple parts like a verb. (strengthens work)
- "Section 4 Notation paragraph" Clarify the relationship of Z and z_i, namely, that z_i is likely the i-th row of Z. Alternatively use Z_{i, \cdot} (strengthens work)
- "Section 4.1" is also not self-contained. namely, what are \mathcal R_{c_1} and \mathcal R_{c_2}. (acceptance)
- "Section 4.2", "efficient for discrete graph structures, and follows:" You probably intended "efficient for discrete graph structures, as follows:" (strengthens work)
- Throughout this work, you use an incorrect command for the indicator function. Consider, for instance, equation 3. Instead of an indicator, you are using a symbol that resembles \mathbb I. (strengthens work)
- "Equation (4)," you did not define p^* (acceptance)
- "Equation (5)," without further context, it is not clear what ||Sigmoid(\Omega)||_1 is. Also, as part of this regularization paragraph, you should refer to the original motivation from GNNInterpreter. (acceptance)
- "Equation (8)," you never specify what \gamma is or how it is chosen. (acceptance)
- "Section 5," "We implement the missing parts and make it public as linked in the abstract." As of now, there is no link (nor a blank link replacement for anonymity) in the abstract. What did you intend here? (strengthens work)
- "Section 5.4," This is more of a question although you could include some more motivation or explanation here: Why do you expect graphs generated from GNNInterpreter to function better, or even differently, than random graphs from the classes when combined via a random edge? (strengthens work)
- "Section 5.5" "target range for the corresponding class pair classes." I do not know what you are trying to say here. In addition, this subsection apparently lacks motivation. While it becomes somewhat clearer what you are attempting after reading section 6.5, it is unclear to me what you are trying to evaluate here. This also complicates the conclusions drawn from section 6.5 and figure 2. (strengthens work)
- "Section 5.6" "To investigate this, we evaluate the boundary statistics (complexity, margin, and thickness) for different target probability ranges and analyze the patterns." This is a very ambiguous statement, and Appendix J (intended to include this evaluation) does not include any text evaluating tables 9-11. I would strongly suggest (acceptance)
- "Section 6.1" Here, you exhibit poor scientific practice. You should first specify the intended confidence, which is likely 95%, then specify the p-value of your evaluation, and then conclude that you reject the negation of claim 1, thus accepting claim 1. (acceptance)
- "Section 6.1" I am having trouble understanding Figure 1. What does a point in this figure represent? How are the success rates computed? (acceptance)
- "Table 1" Please include your proposed GNNInterpreter baseline in Table 1. Also, the chosen precision in Table 1 is higher than necessary. 1e-2 should be sufficient here, giving you more space for the first request. (strengthens work)
- "Figure 2" What are the principal components computed upon? Also, the labels on the x and y-axis should read principal component 1 and principal component 2. Depending on your answer, this might be an unfair evaluation and necessitate additional evaluations using principal components computed on other sets. (acceptance)
- Fix your references. For instance, GNNInterpreter was published at ICLR 2023, and the paper on GAT was published at ICLR 2018, ... . Your references contain 10/23 publications only listed as arXiv, most of which are likely published in some proceedings and should be cited as such.

**Strengths And Weaknesses:**

# Strengths:
- Reproduction is an important avenue of research, especially where groundbreaking work is considered, which "GNNBoundary: Towards Explaining Graph Neural Networks through the Lens of Decision Boundaries" does appear to exhibit.
- Further detailed evaluations are provided, which deepen the investigations into GNNBoundary.
- The clear structure following the main three claims being investigated.

# Weaknesses:
- This work is generally poorly written:
  - This work does not cover related work. Special consideration should be given to the work since the publication of GNNBoundary (also considering about 1 year prior due to submission and recency).
  - This work is not self-contained. There are multiple mentions of variables that are not explained in this work. Specifics follow in the requested changes.
  - There are multiple grammar, spelling, and typo mistakes throughout this work, which, in some cases, lead to poor comprehensibility.
  - Some parts of this work seem to be in poor order or lack well-structured motivation.
  - There are multiple points in this work where terms, statements, and descriptions are used that are not clearly defined. More details in the requested changes.
  - Some references refer to arXiv where proceedings-publications are available. More details in the requested changes.
- There is at least one instance of poor scientific practice. Namely, the p-value of some hypothesis is stated before the authors specify a confidence value, followed by the authors specifying a confidence of 98%--a very unusual chosen confidence. Finally, the author chose to state the acceptance of a claim without specifying that they reject the negation of the claim--which the hypothesis test actually evaluated.

---

> ### Author Response · Authors · 2025-04-02
> **Thanks to the Reviewer & Updates**
>
> Thank you for your detailed and constructive feedback. It helped us strengthen our work and was insightful.
>
> **Writing**: We agree with all of your points and changed the paper accordingly.
>
> **We disagree on the following:**
>
> - Presentation of Table 1: We intend to provide a pure reproduction of Table 1 in Wang & Shen (2024) that allows for a direct comparison. The new baseline is presented in Table 4.
> - **Missing explanation in Section 3.3:** We do provide a full definition and explanation what \mathcal{B}_{c_1 \parallel c_2} is used for: “The decision boundary between class $c_1$ and class $c_2$ is defined as \mathcal{B}_{c_1 \parallel c_2} = \{ G : f{c_1}(G) = f_{c_2}(G) > f_{c'}(G), \forall c' \neq c_1, c_2 \}, representing the set of graphs $G$ where the classifier assigns equal probability to classes $c_1$ and $c_2$, while ranking them higher than all other classes.”
>
> **Answers to your questions:**
>
> - Missing Link in the abstract: We make our full repository public including a completion of the GNNBoundary code, our extensions and the evaluation framework. It is omitted for now to guarantee anonymity. However, we did include the code in the supplementary material.
>
> **Request for review:**
>
> - **Text evaluating Tables 9-11 in Section 6.6:** We added additional evaluation in Section 6.6 regarding Tables 9-11. Are you satisfied with our additions or do you request an even more detailed discussion in the appendix?
>
> **Statistical Significance of Regression:** We agree with your points and adapted Section 6.1. accordingly.
>
> We thank you once more for your valuable feedback that lead to many crucial improvements of our work.

---

> > ### Comment · Reviewer_mFFx · 2025-04-04
> > **Thank you for your changes.**
> >
> > Thank you for responding and making the requested changes for the most part. I can agree with what you have changed and not changed.
> >
> > I have one more change request. Your reference "Degree: Decomposition based explanation for graph neural networks" was published at ICLR 22, but you cite it as an arXiv paper. Please remedy this oversight.

---

> > > ### Author Response · Authors · 2025-04-04
> > > **Thanks to the Reviewer**
> > >
> > > We highly appreciate your attention to detail and posted a new revision where we changed to citing the ICLR proceedings. Thank you for that hint.

---

### Review · Reviewer_mQYL · 2025-03-12

**Summary Of Contributions:**

This paper conducts a reproducibility analysis of the GNNBoundary paper by Wang and Shen. The authors the 3 main claims of Wang and Shen, conducted experiments on the same datasets used by Wang and Shen (and additional datasets), as well as provide additional guidance via a novel baseline and hyperparameter tuning. They discover that while two of the three main claims in Wang and Shen are reproduced, the third main claim is only partially verified due to high variation of results.

**Audience:**

Yes

**Claims And Evidence:**

Yes

**Requested Changes:**

The requested changes below corresponds to weakness 1 and 2 above.

1. Give a paragraph/subsection that A. gives a more comprehensive literature review that positions GNNBoundary within the literature of explainability or GNNs B. give a brief explanation on why GNNBoundary is meaningful/important/interesting to replicate, as opposed to other competitors.

2. Add a section/paragraph (in the supplement/appendix if space is a constraint) that gives a brief outline/notation setup for the types of GNNs that are relevant to this paper.

**Strengths And Weaknesses:**

Strengths:

1. the paper is written in a clear and accessible manner, where the goals and contributions of the paper are clearly stated.
2. the authors provided a comprehensive reproducibility analysis, and thoroughly conducted additional experiments with a new baseline and extra datasets
3. the topic of GNNs, explainability and reproducibility are relevant both to the broader ML community and to the readership of this journal.
4. The materials presented here are, to the best of my knowledge, grounded and technically sound.

Weaknesses:

I state the weaknesses that I perceive below, with corresponding requested changes in the "Requested Changes" section.

1. it is not immediately clear to the reader from the paper of WHY GNNBoundary is chosen to be reproduced, out of all the potentially relevant literature. In other words, what makes GNNBoundary stand out and interesting/important to reproduce? This is important to aid the reader in understanding the context of this paper.

2. it is desirable that the paper be as self-contained as possible. The authors did a good job setting up the notation and framework of GNNBoundary, and I find the background for GNNBoundary and Decision regions satisfactory. However, there is not sufficient background information on GNNs for a general ML audience to follow. For example, in equation (1) notations like H and sigma are directly introduced, with the assumption that the readers know GNNs very well already.

---

> ### Author Response · Authors · 2025-04-02
> **Thanks to the Reviewer & Updates**
>
> Thank you for your well-structured and constructive feedback. It helped us improve the paper:
>
> - 1A: We added a “Related Work” Section (cf. Appendix A)
> - 1B: In addition to the already existing general motivation for explainability research in the introduction, we added a reference to a recent reproducability study (https://openreview.net/pdf?id=8cYcR23WUo) on the highly related GNNInterpreter paper that could not entirely confirm the results. GNNBoundary builds on GNNInterpreter, claims remarkable results and is also the first paper to analyze decision boundaries of GNNs, which motivated us to conduct the work at hand. We added this rationale more explicitly to the introduction (cf. Section 1).
> - 2: Following up on the feedback from the other reviewers, we improved the notation which should improve the readability of the paper for readers without a background in Graph Neural Networks.
>
> We thank you once more for your valuable feedback that lead to several improvements of our work.

---

> > ### Comment · Reviewer_mQYL · 2025-04-30
> > **Reply to Updates**
> >
> > Thank you for the updates. I am satisfied with the updates made. I have no further suggestion/comments.

---

### Review · Reviewer_KrTr · 2025-03-25

**Summary Of Contributions:**

The authors reproduce the findings of *GNNBoundary: Towards Explaining Graph Neural Networks Through the Lens of Decision
Boundaries (Wang & Shen, 2024)*, with additional hyper-parameter searching in the experiments. They further develop an additional baseline using GNNInterpreter graphs, investigate two real-world datasets, and explore the trade-off between boundary metric approximation quality and target class probability ranges.

**Audience:**

Yes

**Claims And Evidence:**

Yes

**Requested Changes:**

* Experiments: The scalability and effectiveness of the method should be validated on larger datasets. Otherwise, the results are less convincing, and the authors should provide complexity analysis and admit this limitation. This is critical to changing my recommendation.

* Writing: A large part of the main contexts is the repetition of previous paper without introducing additional information. This leads to redundancy and makes the contribution of the paper vague. Improving writing would strengthen the work.

* Theory and new conclusions: The paper has no theoretical results and very limited insights, which is crucial to strengthening the work. The authors are also strongly encouraged to include more novel results in additional to validating or reproducing known results.

**Strengths And Weaknesses:**

Strengths: the paper includes extensive experiments, containing both the original findings in previous studies and additional results.

Weaknesses:
* New conclusions from this paper is somewhat inadequate, limiting the originality and significance of the paper.

* For the inconsistent conclusions, the paper does not provide much theoretical explanations nor intuitions/insights.

* All the datasets used in the paper are of small scales. The conclusions may not apply to large-scale datasets and complex GNNs, thus more experiments are required. As admitted in the experimental part of the paper, the method is hard to run even on some moderate size graphs, which makes me concern about the scalability and effectiveness of the method at large scale.

---

> ### Author Response · Authors · 2025-04-02
> **Thanks to the Reviewer & Updates**
>
> We thank you for your feedback and appreciate the opportunity to strengthen our work. We implemented most improvements requested by you and the other reviewers.
>
> **Experiments:** We recognize the importance of scalability and therefore tested GNNBoundary on the Reddit dataset which has a graph size of about 10 times of the other datasets on average (cf. Table 5 in the Appendix and Section 5.2). In Section 6.3, we detailed why we could not reach convergence with such large graphs mentioning how the number of edges scales with the number of nodes. As requested, we added an explicit mentioning of the time and memory complexity of GNNBoundary in the same paragraph. We think that this (now improved) reasoning in combination with the empirical evidence on the Reddit dataset sufficiently proofs that scalability is a major limitation of GNNBoundary. This limitation is also mentioned in the conclusion: “… no results for
> the other (Reddit) due to a deficient scalability with larger graphs.”.
>
> **Writing:** We improved the paper’s writing following the received reviews. In an effort to make the paper as self-contained as possible (as requested also by the other reviewers), we do summarize contents from GNNBoundary in Sections 3 & 4. Following the reviews, we made many small updates to Sections 3 & 4 to make explanations more clear and independent of the readers’ background. While many basic explanations indeed overlap with the Wang & Shen (2024), we often provide an alternative explanation/intuition, also referencing the original papers for the boundary metrics.
>
> **Theory and new conclusions:** Given that this work is a reproducability study, our intentions are
>
> 1. to verify the GNNBoundary authors’ results.
> 2. provide additional guidance for researchers and practitioners how to use / build on GNNboundary.
> 3. and provide additional insights regarding the general applicability of GNNBoundary.
>
> We do so by
>
> - Reproducing all results from Wang & Shen (2024) [point 1]
> - Tuning and analyzing hyperparameters [point 2]
> - Adding more real-world datasets to the benchmark to proof real-world applicability, including an exceptionally large dataset (Reddit) [point 3]
> - Proposing a new random baseline [point 3]
> - Analyzing the robustness of the boundary detection with GNNBoundary [point 3]
> - Proposing a new boundary embedding discovery method underlying the effectiveness of GNNBoundary [point 3]
> - Analyzing the boundary statistics on different target probability ranges [point 1 & 3]
>
> Developing an alternative/competing theoretical framework is clearly out of scope for this work.
>
> We commenced this work under the scope given by the MLRC: https://reproml.org/call_for_papers/ (see “Scope”). Comparing our work to last year’s MLRC proceedings including a reproducabiliity study on GNNInterpreter (https://openreview.net/pdf?id=8cYcR23WUo), we see our work well-aligned with that scope. If you disagree, we would kindly ask you to give specific examples / expectations.
>
> We hope our changes reflect your intentions. Should we have missed something, we kindly ask you to give specific pointers to our work that need improvement.
>
> We thank you once more for your feedback.

---

> > ### Comment · Reviewer_KrTr · 2025-04-14
> >
> > Thank you for your clarifications. I recognize the changes made by the authors and agree with the authors' claims regarding the scope of the paper. However, it should be admitted that limited new results would down-weight the contribution of the paper to some extent.

---

### Decision · Action_Editor_DBsK · 2025-05-27

**Recommendation:** Accept as is

**Comment:**

This is a reproducibility study. The authors took some effort into hyperparameter tuning and evaluating new datasets, providing some new insights on which datasets the reproduced method still works on. Hence, although borderline, I would recommend acceptance, following most of the reviewers' recommendations.

**Audience:**

Researchers interested in applying GNNBoundary will benefit from the hyperparameter tuning efforts of the authors. Hence, it is interesting to some researchers.

**Claims And Evidence:**

All claims are sufficiently supported. Most of the reviewers endorsed the acceptance of the present work and stated that the requested changes were adequately addressed.